# The structural repertoire of *Fusarium oxysporum* f. sp. *lycopersici* effectors revealed by experimental and computational studies

**Daniel S Yu[1], Megan A Outram[1]\*[†], Ashley Smith[1], Carl L McCombe[1], Pravin B Khambalkar[1], Sharmin A Rima[1], Xizhe Sun[1,2], Lisong Ma[1,3], Daniel J Ericsson[1,4], David A Jones[1], Simon J Williams[1]\***

[1]Research School of Biology, The Australian National University, Canberra, Australia; [2]Key Laboratory of Hebei Province for Plant Physiology and Molecular Pathology, College of Life Sciences, Hebei Agriculture University, Baoding, China; [3]State Key Laboratory of North China Crop Improvement and Regulation, College of Horticulture, Hebei Agricultural University, Baoding, China; [4]The Australian Nuclear Science and Technology Organisation, Australian Synchrotron, Clayton, Australia

**\*For correspondence:**
megan.outram@csiro.au (MAO);
simon.williams@anu.edu.au (SJW)

**Present address:** [†]Black Mountain Science and Innovation Park, CSIRO Agriculture and Food, Canberra, Australia

**Competing interest:** The authors declare that no competing interests exist.

**Abstract** Plant pathogens secrete proteins, known as effectors, that function in the apoplast or inside plant cells to promote virulence. Effector recognition by cell-surface or cytosolic receptors results in the activation of defence pathways and plant immunity. Despite their importance, our general understanding of fungal effector function and recognition by immunity receptors remains poor. One complication often associated with effectors is their high sequence diversity and lack of identifiable sequence motifs precluding prediction of structure or function. In recent years, several studies have demonstrated that fungal effectors can be grouped into structural classes, despite significant sequence variation and existence across taxonomic groups. Using protein X-ray crystallography, we identify a new structural class of effectors hidden within the secreted in xylem (SIX) effectors from *Fusarium oxysporum* f. sp. *lycopersici* (*Fol*). The recognised effectors Avr1 (SIX4) and Avr3 (SIX1) represent the founding members of the *Fol* dual-domain (FOLD) effector class, with members containing two distinct domains. Using AlphaFold2, we predicted the full SIX effector repertoire of *Fol* and show that SIX6 and SIX13 are also FOLD effectors, which we validated experimentally for SIX6. Based on structural prediction and comparisons, we show that FOLD effectors are present within three divisions of fungi and are expanded in pathogens and symbionts. Further structural comparisons demonstrate that *Fol* secretes effectors that adopt a limited number of structural folds during infection of tomato. This analysis also revealed a structural relationship between transcriptionally co-regulated effector pairs. We make use of the Avr1 structure to understand its recognition by the I receptor, which leads to disease resistance in tomato. This study represents an important advance in our understanding of *Fol*-tomato, and by extension plant–fungal interactions, which will assist in the development of novel control and engineering strategies to combat plant pathogens.

## eLife assessment

This study provides **important** new insights into the structural diversity of effectors – proteins secreted by pathogens and symbionts into host cells – from the plant-associated fungus *Fusarium oxysporum* f. sp. *lycopersici*. The study provides a **convincing** approach to elucidate how effectors

navigate their host environment by exploiting both computational and experimental approaches to understand how their structure influences binding partners. The work will be of interest to those studying molecular host–microbe interactions and disease protection.

## Introduction

*Fusarium oxysporum* is a soil-borne fungal pathogen responsible for destructive vascular wilt diseases in a wide range of plants. It ranks within the top 10 important fungal pathogens in terms of scientific and economic importance (*Dean et al., 2012*). The best-characterised *F. oxysporum* pathosystem involves *F. oxysporum* f. sp. *lycopersici* (*Fol*) and tomato. Previous studies of *Fol*-infected tomato identified a number of fungal proteins within the xylem sap (*Rep, 2005*). These secreted in xylem (SIX) effector proteins represent major pathogenicity determinants across different *formae speciales* of *F. oxysporum*. Currently, 14 SIX effectors have been identified in *Fol* consisting of small (less than 300 amino acids in length), secreted, cysteine-rich proteins (*Houterman et al., 2007*; *Ma et al., 2010*; *Rep et al., 2004*; *Schmidt et al., 2013*). Most SIX effectors are encoded on the conditionally dispensable chromosome 14 required for *Fol* pathogenicity (*Vlaardingerbroek et al., 2016*). This dispensable chromosome can be horizontally transferred from *Fol* to a non-pathogenic strain of *F. oxysporum*, resulting in a transfer of pathogenicity (*Ma et al., 2010*; *Vlaardingerbroek et al., 2016*). To date, all 14 SIX effectors lack sequence identity with proteins of known function, preventing prediction of function based on their amino acid sequence. Several SIX effectors have been shown to be essential for full virulence, including SIX1, SIX2, SIX3, SIX5, and SIX6 from *Fol* (*Rep et al., 2004*; *Gawehns et al., 2014*; *Ma et al., 2015*; *van der Does et al., 2008*; *Gawehns et al., 2015*), SIX1 from *F. oxysporum* f. sp. *conglutinans* (*Focn*), which infects cabbage (*Li et al., 2016*), SIX4 from *F. oxysporum* isolate Fo5176, which infects *Arabidopsis* (*Thatcher et al., 2012*), and SIX1 and SIX8 from *F. oxysporum* f. sp. *cubense*, which infects banana (*An et al., 2019*; *Widinugraheni et al., 2018*). *Fol SIX3* (*Avr2*) and *SIX5* are adjacent, divergently transcribed genes with a common promoter, and SIX5 has been shown to interact with SIX3 to promote virulence by enabling symplastic movement of SIX3 via plasmodesmata (*Cao et al., 2018*). *Focn SIX8* and *PSE1* (pair with *SIX8* 1) are also a divergently transcribed effector gene pair that function together to suppress phytoalexin production and plant immunity in *Arabidopsis* (*Ayukawa et al., 2021*). In *Fol*, *SIX8* forms a similar gene pair with *PSL1* (*PSE1*-like 1) (*Ayukawa et al., 2021*). Despite their roles in fungal pathogenicity, the virulence functions of most SIX effectors remain unknown.

To combat pathogen attack, plants possess resistance genes that encode immunity receptors capable of recognising specific effectors leading to disease resistance. Four resistance genes, introgressed into tomato from related wild species, have been cloned. *I* and *I-7* encode transmembrane receptor proteins containing extracellular leucine-rich repeat (LRR) domains and short cytoplasmic domains (LRR-RPs) (*Gonzalez-Cendales et al., 2016*; *Catanzariti et al., 2017*). *I-2* encodes a cytoplasmic receptor containing nucleotide binding (NB) and C-terminal LRR domains (*Simons et al., 1998*), while *I-3* encodes a transmembrane protein with an extracellular S-receptor-like domain and cytoplasmic serine/threonine kinase domain (SRLK) (*Catanzariti et al., 2015*). *Fol* Avr1 (SIX4), Avr2 (SIX3), and Avr3 (SIX1) are recognised by tomato immunity receptors, I, I-2, and I-3, respectively, leading to effector-triggered immunity (ETI) and disease resistance (*Rep et al., 2004*; *Houterman et al., 2008*; *Houterman et al., 2009*).

By understanding the function of *F. oxysporum* effector proteins, and how specific effectors are detected by immunity receptors, we (and others) hope to develop novel disease management strategies targeting vascular wilt diseases. Protein structure studies of effectors provide one avenue to assist this pursuit. Currently, Avr2 represents the only SIX effector whose protein structure has been determined (*Di et al., 2017*). Interestingly, the β-sandwich fold of Avr2 revealed that this effector shares structural homology to ToxA from *Pyrenophora tritici-repentis* and AvrL567 from *Melampsora lini* (*Sarma et al., 2005*; *Wang et al., 2007*), despite a lack of sequence identity. The observation of structural classes for effectors without identifiable domains or homologies to proteins of known function has been demonstrated experimentally for four effector structural families, including the so-called MAX (*M*agnaporthe oryzae *A*vr effectors and To*x*B from *P. tritici-repentis*) (*de Guillen et al., 2015*), RALPH (*R*N*A*se-*L*ike *P*roteins associated with *H*austoria) (*Spanu, 2017*), LARS (*L*eptosphaeria

**A**virulence and **S**uppressing) (*Lazar et al., 2022*), and ToxA-like families (*Di et al., 2017*; *Sarma et al., 2005*; *Wang et al., 2007*).

Combining experimental and computational approaches, we present the structural repertoire of sequence unrelated effectors utilised by *Fol* during infection of tomato, including the classification of a new effector family, the FOLD (***Fol d***ual-domain) effectors. We show using structural comparisons that FOLD effectors are widely distributed in phytopathogenic fungi as well as symbionts. Further, we define the domains and residue that mediate the recognition of the FOLD effector, Avr1, by its corresponding immunity receptor.

## Results

### The structures of Avr1 and Avr3 adopt a similar dual-domain fold

Avr1 and Avr3 are cysteine-rich effectors that belong to the K2PP (Kex2-processed pro-domain) effector class (*Outram et al., 2021b*; *Outram et al., 2021a*). To help understand their function, and recognition by I and I-3, we sought to solve their structures using X-ray crystallography. Using our optimised protein production strategy (*Yu et al., 2021*), we produced Avr1 (Avr1$^{18-242}$) and Avr3 (Avr3$^{22-284}$) in *Escherichia coli* for crystallisation studies (*Figure 1—figure supplement 1A and B*). Crystals were obtained for Avr3$^{22-284}$ (hereafter referred to as Avr3) (*Figure 1—figure supplement 1B*); however, Avr1$^{18-242}$ failed to crystallise. Previously, we demonstrated that pro-domain removal from the K2PP effector SnTox3 was required to obtain protein crystals (*Outram et al., 2021b*) and predicted this may also be important for Avr1. Treatment of Avr1$^{18-242}$ with Kex2 in vitro resulted in a predominant Avr1 band of ~20 kDa consistent with a mature Avr1$^{59-242}$ protein; however, lower molecular weight bands were also observed, suggesting in vitro Kex2 cleavage at additional sites (*Outram et al., 2021b*). To address this, Avr1 was engineered with an internal thrombin cleavage site (replacing the Kex2 site) to produce a single Avr1$^{59-242}$ product after thrombin cleavage (hereafter referred to as Avr1). This protein was subsequently used for crystallisation studies, resulting in rectangular plate-like crystals (*Figure 1—figure supplement 1A*).

The crystal structures of Avr1 and Avr3 were solved using a bromide-ion-based single-wavelength anomalous diffraction (SAD) approach (*Supplementary file 1*) and subsequently refined using a native dataset to a resolution of 1.65 Å and 1.68 Å, respectively (*Figure 1A and B*). Despite sharing low amino-acid sequence identity (19.5%), Avr1 and Avr3 adopt a structurally similar dual-domain protein fold. Interpretable, continuous electron density was observed from residue 96 in Avr3 and some regions of the intact pro-domain could be interpreted in the electron density (residues 26–49) (*Figure 1—figure supplement 2A*). We also identified regions of the pro-domain (residues 23–45) of Avr1 in the electron density, despite thrombin cleavage of the pro-domain prior to crystallisation (*Figure 1—figure supplement 1A*). This indicates that an association between respective Avr and pro-domain was maintained post cleavage in vitro (*Figure 1—figure supplement 2B*). The importance of this association, if any, remains unclear, but for simplicity, the pro-domains were excluded from further analysis.

The Avr1 and Avr3 N-terminal domain (N-domain), consisting of an N-terminal α-helix followed by five β-strands, and C-terminal domain (C-domain), consisting of a β-sandwich architecture, involving seven or eight β-strands are very similar with a root-mean-square deviation (RMSD) of 2.1 Å and 2.8 Å, respectively (superposition performed using DALI server; *Holm, 2022*; *Figure 1*). While the individual domains are very similar, superposition of the dual-domain structures returns an RMSD of ~3.4 Å. The larger difference is due to a rotation between the N- and C-domains (*Figure 1E*). The structures of Avr1 and Avr3, when compared with the solved structures of other fungal effectors, demonstrate that they adopt a unique two-domain fold and represent the founding members of a new structural class of fungal effectors we have designated the FOLD effectors.

### SIX6 and SIX13 belong to the FOLD effector family

We were interested in determining whether the other SIX effectors belonged to the FOLD effector family. One conserved sequence feature observed in Avr1 and Avr3 was the spacing of the six cysteines within the N-domain. We analysed the cysteine spacing of the other SIX effectors and found that SIX6 and SIX13 contained a cysteine profile like Avr1 and Avr3 (*Figure 2A*), suggesting that they may be FOLD effectors. With the recent advances in ab initio structural prediction by Google DeepMind's

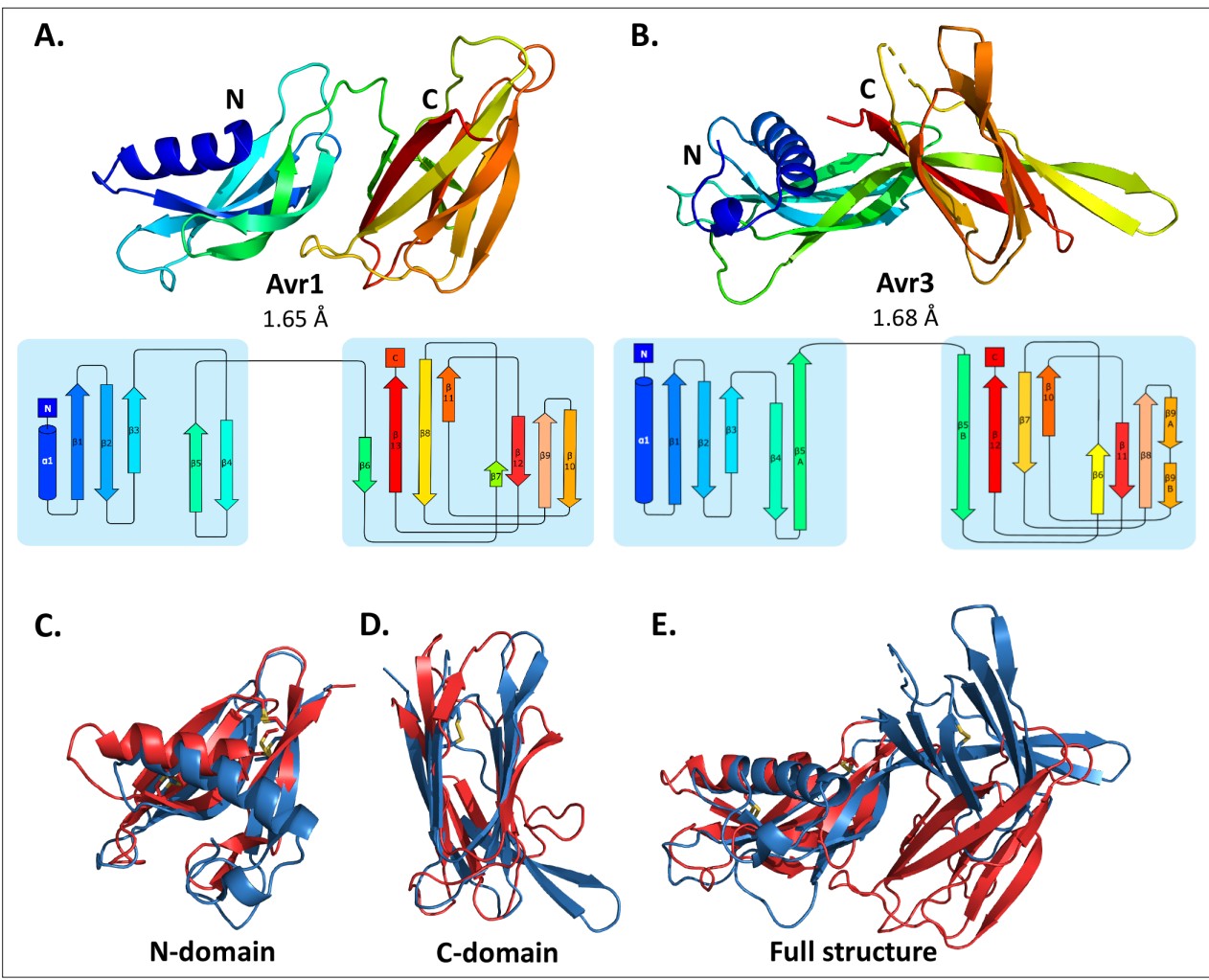

**Figure 1.** Crystal structures of Avr1 and Avr3 from *Fol* adopt a similar structural fold. Ribbon diagrams of (**A**) Avr1 and (**B**) Avr3 coloured from N- (blue) to C-terminus (red) showing the dual-domain structural fold (top panels) and secondary structure topology map (bottom panels) of Avr1 and Avr3, respectively. For both, the N-domain is shown on the left and the C-domain is shown on the right. The colours of the secondary structural elements match the colours depicted on the crystal structure. Structural alignments of Avr1 (shown in red) and Avr3 (shown in blue) showing (**C**) N-domains alone, (**D**) C-domains alone, and (**E**) full structures. Disulfide bonds are shown in yellow. Structural alignment was performed using the pairwise alignment function of the DALI server (**Holm, 2022**).

The online version of this article includes the following source data and figure supplement(s) for figure 1:

**Figure supplement 1.** Crystallisation of Avr1, Avr3, SIX6, SIX8, SIX13, and PSL1.

**Figure supplement 1—source data 1.** Unedited and uncropped SDS-PAGE gel for *Figure 1—figure supplement 1A*, Avr1[18-242].

**Figure supplement 1—source data 2.** Unedited and uncropped SDS-PAGE gel for *Figure 1—figure supplement 1A*, Avr1[18-242], with relevant bands labelled.

**Figure supplement 1—source data 3.** Unedited and uncropped SDS-PAGE gel for *Figure 1—figure supplement 1A*, Avr1[59-242].

**Figure supplement 1—source data 4.** Unedited and uncropped SDS-PAGE gel for *Figure 1—figure supplement 1A*, Avr1[59-242], with relevant bands labelled.

**Figure supplement 1—source data 5.** Unedited and uncropped SDS-PAGE gel for *Figure 1—figure supplement 1B*, Avr3[22-284].

**Figure supplement 1—source data 6.** Unedited and uncropped SDS-PAGE gel for *Figure 1—figure supplement 1B*, Avr3[22-284], with relevant bands labelled.

**Figure supplement 1—source data 7.** Unedited and uncropped SDS-PAGE gel for *Figure 1—figure supplement 1C*, SIX6[17-225].

**Figure supplement 1—source data 8.** Unedited and uncropped SDS-PAGE gel for *Figure 1—figure supplement 1C*, SIX6[17-225], with relevant bands labelled.

**Figure supplement 1—source data 9.** Unedited and uncropped SDS-PAGE gel for *Figure 1—figure supplement 1C*, SIX6[58-225].

*Figure 1 continued on next page*

*Figure 1 continued*

**Figure supplement 1—source data 10.** Unedited and uncropped SDS-PAGE gel for *Figure 1—figure supplement 1C*, SIX6[58-225], with relevant bands labelled.

**Figure supplement 1—source data 11.** Unedited and uncropped SDS-PAGE gel for *Figure 1—figure supplement 1D*, SIX13[22-293].

**Figure supplement 1—source data 12.** Unedited and uncropped SDS-PAGE gel for *Figure 1—figure supplement 1D*, SIX13[22-293], with relevant bands labelled.

**Figure supplement 1—source data 13.** Unedited and uncropped SDS-PAGE gel for *Figure 1—figure supplement 1E*, SIX8[19-141].

**Figure supplement 1—source data 14.** Unedited and uncropped SDS-PAGE gel for *Figure 1—figure supplement 1E*, SIX8[19-141], with relevant bands labelled.

**Figure supplement 1—source data 15.** Unedited and uncropped SDS-PAGE gel for *Figure 1—figure supplement 1E*, SIX8[50-141].

**Figure supplement 1—source data 16.** Unedited and uncropped SDS-PAGE gel for *Figure 1—figure supplement 1E*, SIX8[50-141], with relevant bands labelled.

**Figure supplement 1—source data 17.** Unedited and uncropped SDS-PAGE gel for *Figure 1—figure supplement 1F*, PSL1[18-111].

**Figure supplement 1—source data 18.** Unedited and uncropped SDS-PAGE gel for *Figure 1—figure supplement 1F*, PSL1[18-111], with relevant bands labelled.

**Figure supplement 2.** Continuous electron density of the pro-domain is present in the crystal structures of Avr1, Avr3, and SIX6.

**Figure supplement 3.** Circular dichroism (CD) analysis of purified recombinant proteins.

AlphaFold2 (*Jumper et al., 2021*), we predicted the structures of the SIX effectors to determine whether, as suggested by our sequence analysis, the other SIX effectors are FOLD effector family members.

As an initial step we benchmarked the AlphaFold2-predicted models of Avr1 and Avr3 downstream of the Kex2 cleavage site (Avr1[59-242] and Avr3[96-284]) against our experimentally determined structures (*Figure 2—figure supplement 1*). The AlphaFold2 model of Avr1 returned a low average per-residue confidence score (pLDDT = 55%) and the RMSD was 6.9 Å when model and structure were compared; however, the dual domain architecture was correctly predicted with a Z-score of 11.3 identified using a DALI pairwise structural comparison (*Figure 2—figure supplement 1A and E*). The AlphaFold2 model of Avr3 returned a high average pLDDT score (92%) and superimposed well to the solved structure (*Figure 2—figure supplement 1B*), despite a slight skew between the orientation of the individual domains (RMSD = 3.6 Å overall; 1.1 Å for the N-domain; 0.8 Å for the C-domain). This demonstrated that accurate FOLD effector prediction was possible using AlphaFold2.

We subsequently generated SIX6 and SIX13 models, downstream of the predicted Kex2 cleavage site (SIX6[58-225], SIX13[78-293]), using AlphaFold2 and obtained high average confidence scored models supporting their inclusion in the FOLD family (*Figure 2—figure supplement 2*). To validate this experimentally, we produced SIX6[58-225] and SIX13[22-293] (hereafter referred to as SIX6 and SIX13) as described for Avr1/Avr3 and obtained crystals for both proteins (*Figure 1—figure supplement 1*). While the SIX13 crystals diffracted poorly, the SIX6 crystals diffracted X-rays to ~1.9 Å, and we solved the structure of SIX6 using the AlphaFold2-generated model as a template for molecular replacement (*Figure 2B*, *Supplementary file 1*), confirming its inclusion as a member of the FOLD family. Despite lacking an N-terminal helix, the N-domain contains five β-strands held together by three disulfide bonds with an arrangement, identical to Avr1 and Avr3. The C-domain is an eight stranded β-sandwich that is stabilised by a single disulfide bond (unique to SIX6 compared to Avr1 and Avr3) connecting the β7 and β12 strands. Like Avr1, we identified regions of the pro-domain within the SIX6 structure (residues 29–46), despite cleavage of the pro-domain prior to crystallisation (*Figure 1—figure supplement 2C*), but only within one molecule in the asymmetric unit (*Figure 1—figure supplement 2D*). For structural analysis, we used the structured region of Chain A of SIX6 (*Figure 2B*).

## FOLD effectors are distributed across multiple fungal genera

Despite structural similarities, the FOLD effectors are divergent in their amino acid sequences, sharing 15.5–22.5% sequence identities between all members (*Figure 2A*). Homologues of FOLD effectors are dispersed across multiple *formae speciales* of *F. oxysporum* (*Figure 2—figure supplement 3*; *Schmidt et al., 2013*; *Gawehns et al., 2014*; *Batson et al., 2021*; *Czislowski et al., 2018*; *Lievens et al., 2009*; *van Dam et al., 2016*). We were interested in understanding the distribution of FOLD

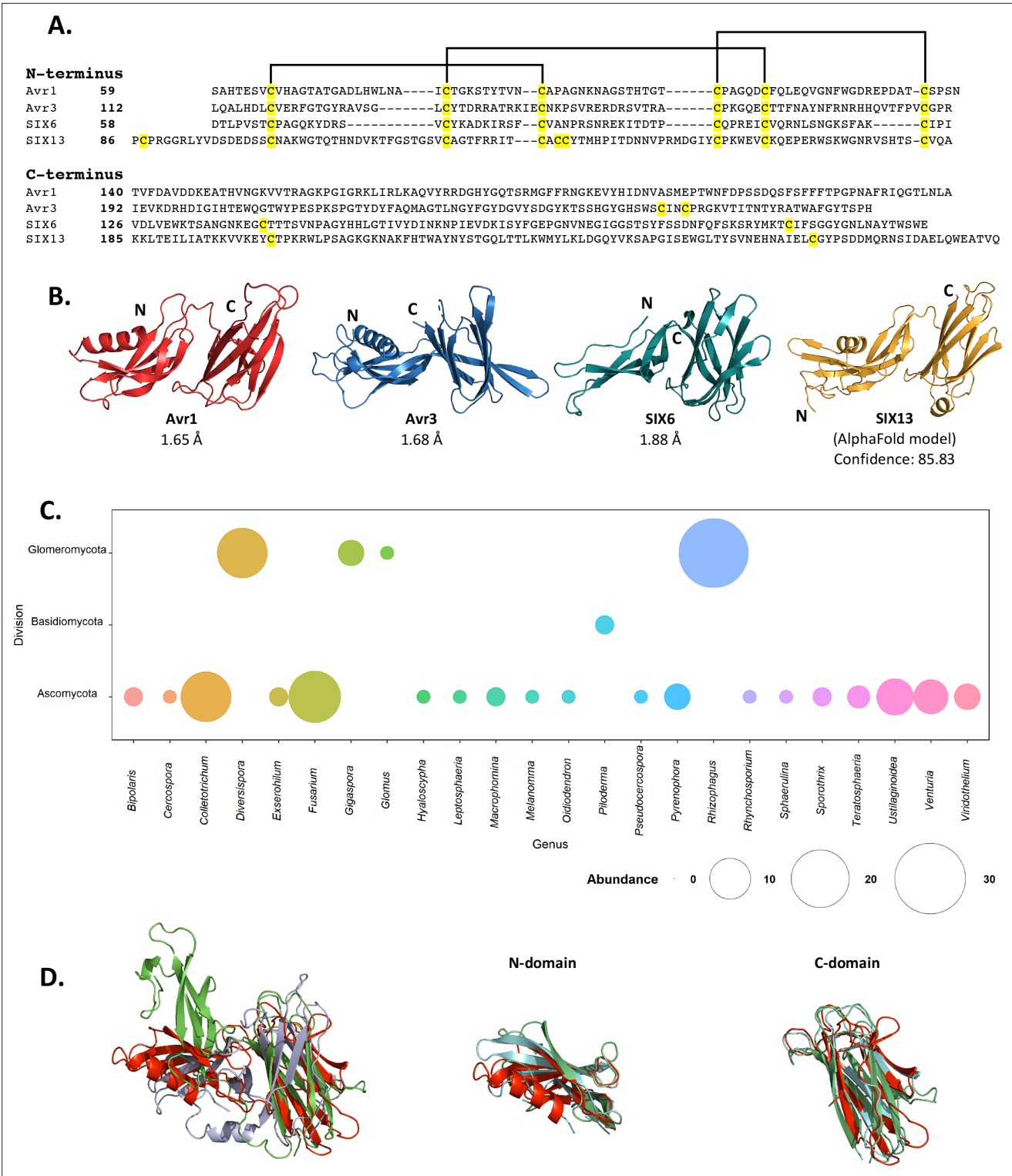

**A.**

**N-terminus**

| Avr1 | 59 | SAHTESV**C**VHAGTATGADLHWLNA----I**C**TGKSTYTVN--**C**APAGNKNAGSTHTGT------**C**PAGQD**C**FQLEQVGNFWGDREPDAT-**C**SPSN |
| Avr3 | 112 | LQALHDL**C**VERFGTGYRAVSG-------L**C**YTDRRATRKIE**C**NKPSVRERDRSVTRA-----**C**PKGQE**C**TTFNAYNFRNRHHQVTFPV**C**GPR |
| SIX6 | 58 | DTLPVST**C**PAGQKYDRS----------V**C**YKADKIRSF--**C**VANPRSNREKITDTP------**C**QPREI**C**VQRNLSNGKSFAK------**C**IPI |
| SIX13 | 86 | P**C**PRGGRLYVDSDEDSS**C**NAKWGTQTHNDVKTFGSTGSV**C**AGTFRRIT---**C**AC**C**YTMHPITDNNVPRMDGIY**C**PKWEV**C**KQEPERWSKWGNRVSHTS-**C**VQA |

**C-terminus**

| Avr1 | 140 | TVFDAVDDKEATHVNGKVVTRAGKPGIGRKLIRLKAQVYRRDGHYGQTSRMGFFRNGKEVYHIDNVASMEPTWNFDPSSDQSFSFFFTPGPNAFRIQGTLNLA |
| Avr3 | 192 | IEVKDRHDIGIHTEWQGTWYPESPKSPGTYDYFAQMAGTLNGYFGYDGVYSDGYKTSSHGYGHSWS**C**IN**C**PRGKVTITNTYRATWAFGYTSPH |
| SIX6 | 126 | VDLVEWKTSANGNKEG**C**TTTSVNPAGYHHLGTIVYDINKNPIEVDKISYFGEPGNVNEGIGGSTSYFSSDNFQFSKSRYMKT**C**IFSGGYGNLNAYTWSWE |
| SIX13 | 185 | KKLTEILIATKKVVKEY**C**TPKRWLPSAGKGKNAKFHTWAYNYSTGQLTTLKWMYLKLDGQYVKSAPGISEWGLTYSVNEHNAIEL**C**GYPSDDMQRNSIDAELQWEATVQ |

**B.**

Avr1  1.65 Å

Avr3  1.68 Å

SIX6  1.88 Å

SIX13 (AlphaFold model) Confidence: 85.83

**C.**

**D.**

N-domain

C-domain

**Figure 2.** *Fol* dual-domain (FOLD) effector family is distributed within *Fusarium oxysporum* and other fungi. (**A**) Amino acid sequence alignment of the mature Avr1, Avr3, SIX6, and SIX13 sequences shows a common cysteine spacing at the N-terminus. The alignment is split into the N-terminus (N-domain; top panel) and C-terminus (C-domain; bottom panel). Cysteine residues are highlighted in yellow, with common disulfide bonding connectivity, as determined by the crystal structures of Avr1 and Avr3, shown with black lines. (**B**) Ribbon diagrams of the Avr1, Avr3, SIX6 crystal structures and SIX13 AlphaFold2 model show a conserved dual-domain structure. The N- and C-termini are labelled. (**C**) Structure-guided search for putative FOLD effectors across fungi using Foldseek webserver. The size of the circles represents abundance with genus. (**D**) Superposition (structural alignment) of representative putative FOLD effectors from the divisions Glomeromycota and Basidiomycota with Avr1 in ribbon representation. Putative FOLD protein

*Figure 2 continued on next page*

*Figure 2 continued*

from *Rhizophagus clarus* (UniProt: A0A2Z6QDJ0) in light blue, and *Piloderma croceum* (UniProt: A0A0C3C2B2) in green. FOLD structural alignment (right), N-domain only (middle), and C-domain only (right).

The online version of this article includes the following source data and figure supplement(s) for figure 2:

**Figure supplement 1.** Comparison of AlphaFold2 models against the experimentally solved structures of Avr1, Avr3, SIX6, and SIX8.

**Figure supplement 2.** Structural alignments of SIX6 and SIX13 with Avr1.

**Figure supplement 3.** Homologues of *Fol* **d**ual-domain (FOLD) effectors are dispersed across multiple *formae speciales* of *F. oxysporum*.

**Figure supplement 3—source data 1.** PDF version of *Figure 2—figure supplement 3*.

effectors in fungi. Previous structural-based searches performed on effector candidates from *Venturia inaequalis* using Avr1 and Avr3 as templates (which we provided to the authors) found three candidates predicted to be FOLD effectors (*Rocafort et al., 2022*). Here, we utilised our experimentally determined structures (Avr1, Avr3, and SIX6) to search for other fungal FOLD effectors within the AlphaFold2 protein structure database (*Varadi et al., 2022*; https://alphafold.ebi.ac.uk/) using the Foldseek webserver (*van Kempen et al., 2023*). This analysis identified 124 putative FOLD protein family members across three divisions of fungi (Ascomycota, Basidiomycota, and Glomeromycota) (*Figure 2C*). Over half of these were found in Ascomycota fungi (73), with expanded families in species of *Fusarium* and *Colletotrichum* (*Figure 2C*, *Supplementary file 2*). Expanded families of FOLD proteins were also observed in the division Glomeromycota that form arbuscular mycorrhiza in plant roots, while two putative FOLD effectors were also predicted in the ectomycorrhizal fungus *Piloderma olivaceum* (division Basidiomycota), which forms mutualistic associations with conifer and hardwood species (*van Kempen et al., 2023*). Structural superposition of members from the three divisions confirms the structural similarities between the N and C domains and highlights that the major differences identified are the orientation of the domains relative to each other (*Figure 2D*), consistent with our experimental data for Avr1, Avr3 and SIX6.

## Distinct structural families exist among the other SIX effectors

With the successful utilisation of AlphaFold2 as a model for molecular replacement (SIX6 structure), and structural similarity searches for FOLD effectors, we decided to perform structural comparisons with the remaining SIX effectors. AlphaFold2 modelling of the effectors was conducted on sequences with the signal peptide and putative pro-domain (if present) removed (*Figure 3—figure supplement 1*). The models and experimentally determined SIX effector structures (Avr1, Avr2, Avr3, and SIX6) were compared using the DALI server (*Holm, 2022*) and a Z-score with a cutoff of >2 was used to indicate structure similarity.

The observed structural similarity between the FOLD effectors was high, with Z-scores above 8 for all comparisons (*Figure 3A*). Avr2, a member of the ToxA-like effector family, exhibited structural similarity with the SIX7 and SIX8 models (Z-scores > 5) (*Figure 3A*). Analysis of the models and topology show that SIX7 and SIX8 both consist of a β-sandwich fold, strongly indicating their inclusion within the ToxA-like structural family (*Figure 3C*, *Figure 3—figure supplement 2*).

Beyond these described structural families, the Z-scores indicated that two additional, but not yet characterised, structural families exist within the SIX effectors. Here, we define these as structural family 3 and 4, consisting of SIX9 and SIX11, and SIX5 and SIX14, respectively (*Figure 3D and E*). The models of SIX9 and SIX11 both consist of five β-strands and either two or three α-helices (*Figure 3D*, *Figure 3—figure supplement 3*), despite sharing only 14% sequence identity. To further our understanding of the putative function of this family, we did a structural search against the protein databank (PDB) and found that both structures share structural similarity to various RNA-binding proteins (Z-scores > 2.5), such as the RNA recognition motif (RRM) fold of the Musashi-1 RNA-binding domain (PDB code: 5X3Z) (*Iwaoka et al., 2017*) with Z-scores of 2.6 and 4.5 for SIX9 and SIX11, respectively.

SIX5 and SIX14 also share limited sequence identity (23%) but the structural predictions show a similar secondary-structure topology consisting of 2 α-helices and 4–6 β-strands (*Figure 3E*, *Figure 3—figure supplement 3*). We compared the models of SIX5 and SIX14 against the PDB using DALI and identified structural similarity toward the *Ustilago maydis* and *Zymoseptoria tritici* KP6 effector (PDB codes: 4GVB and 6QPK) (*Allen et al., 2013*), suggesting that SIX5 and SIX14 belong to the KP6-like

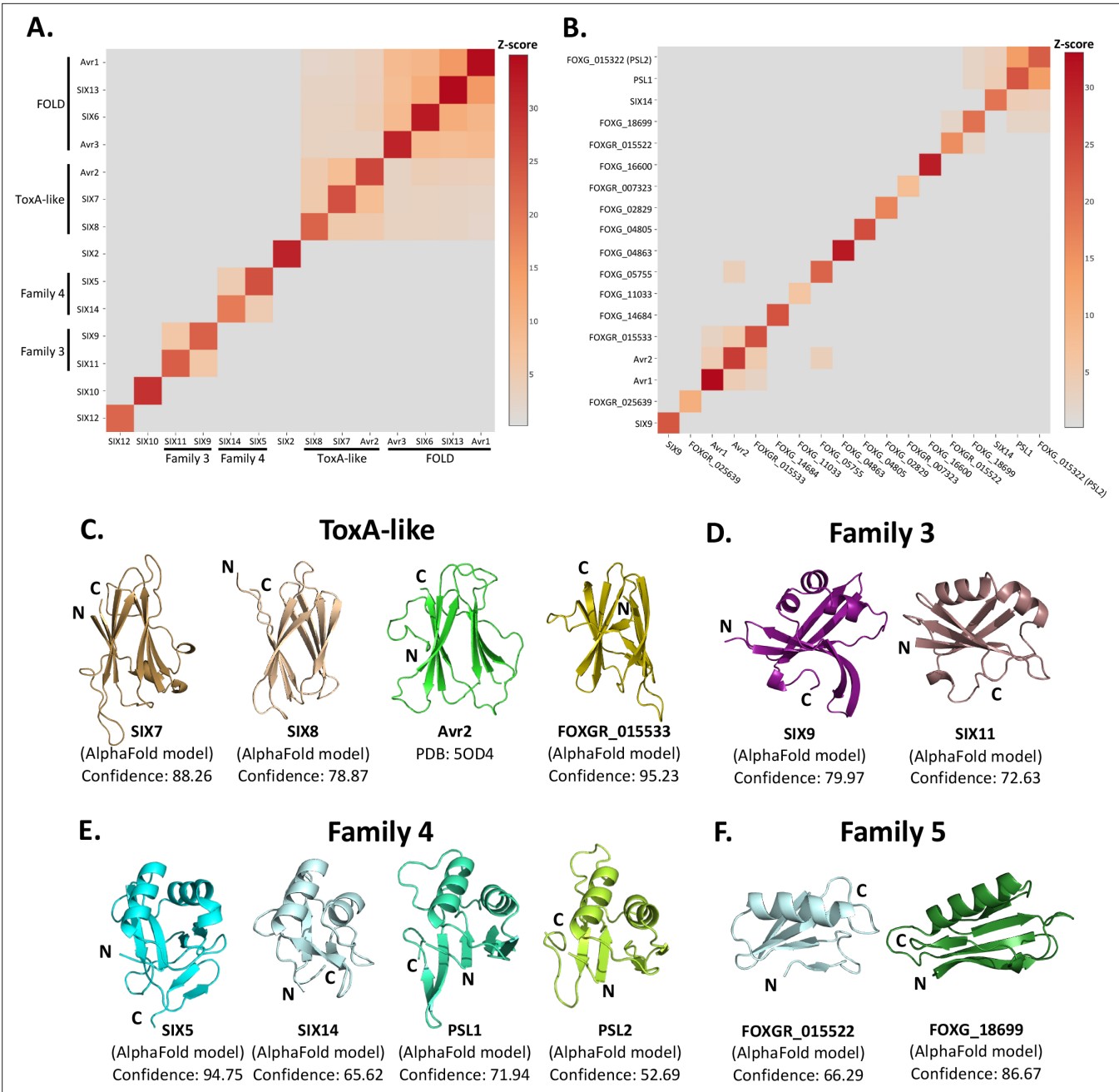

**Figure 3.** Identification of new putative structural families within the SIX effectors. Heat maps showing the structural similarity of structures and AlphaFold2 models of the (**A**) SIX effectors and (**B**) effector candidates from *Fol* in a structural pairwise alignment. Amino acid boundaries that were modelled for each protein are provided in ***Supplementary file 3***. Structural similarity was measured with Z-scores. A cutoff Z-score of 2 was applied for defining structural families. Z-score scale is shown in grey to red spectrum. (**C**) Cartoon representation of the ToxA-like effectors from *Fol*. AlphaFold2 models of SIX7, SIX8, and FOXGR_015533 effector candidate are putative members of the ToxA-like effector family. The crystal structure of Avr2 (***Di et al., 2017***), another member of the ToxA-like effector family, is shown in green for comparison. Cartoon representations of (**D**) family 3, (**E**) family 4, and (**F**) family 5 consisting of members that are predicted to be structurally similar. The N- and C-termini are labelled. Structural similarity searches were performed using the DALI server (***Holm, 2022***).

The online version of this article includes the following figure supplement(s) for figure 3:

**Figure supplement 1.** AlphaFold2 models of all SIX effectors and effector candidates.

**Figure supplement 2.** Structural similarity of SIX effectors against representative solved effector structures from known structural families.

**Figure supplement 3.** Secondary structure topology maps of the representative SIX structural family members.

structural family (*Figure 3—figure supplement 2*). Collectively, this analysis demonstrates that 11 of the 14 SIX effectors group into four different structural families.

## Structural modelling and comparison of an expanded set of *Fol* effectors

The SIX effectors are only a subset of effectors utilised by *Fol* during infection of tomato. Recently, the *Fol* genome was resequenced (*Li et al., 2020*) and reannotated in combination with RNAseq data from *Fol*-infected tomato plants (*Sun et al., 2022*). A total of 26 genes encoding novel effector candidates were identified that were consistently upregulated during *Fol* infection (*Sun et al., 2022*), which were not previously predicted or predicted incorrectly in the original genome annotation (*Ma et al., 2010*). Of these, 14 genes encoded proteins with no recognised domains or motifs based on their amino acid sequences. We generated structural models using AlphaFold2 of these 14 (*Supplementary file 3*, *Figure 3—figure supplement 1*) and structurally aligning them using DALI against SIX effector representatives from each family to assess whether they fell into any of the established families (*Figure 3B*). We found that the predicted structure of FOXGR_015533 adopts a nine β-stranded sandwich and is likely a member of the ToxA-like class (*Figure 3C*). PSL1 (*Ayukawa et al., 2021*) and FOXGR_015322, here designated PSL2, are sequence-related effectors (~85% sequence identity) and show a conserved structure (*Figure 3E*). Both have Z-scores of >2 against family 4 and are likely members of this family.

Based on this analysis, we also suggest an additional structural family. FOXG_18699 and FOXGR_015522 are structurally related (Z-score of 2.2) with a sequence identity of ~29%. While FOXGR_015522 does share some resemblance to family 4, based on manual alignment (*Figure 3F*) and domain topology analysis (*Figure 3—figure supplement 3*), these effectors appear to belong to an independent structural family, designated family 5. Collectively, these data demonstrate that *Fol* utilises multiple structurally related, sequence-diverse, effectors during infection of tomato.

## Interaction between effector pairs from two structural families

In *Fol*, *Avr2* and *SIX5*, and *SIX8* and *PSL1* form a similar head-to-head relationship in the genome with shared promoters and are divergently-transcribed (*Figure 4A*; *Cao et al., 2018*; *Ayukawa et al., 2021*). Previously, studies concerning Avr2 and SIX5 have demonstrated that the proteins function together and interact directly via yeast-two-hybrid analysis (*Ma et al., 2015*). Homologues of SIX8 and PSL1 from *Focn* (SIX8 and PSE1) are also functionally dependent on each other; however, an interaction could not be established in yeast (*Ayukawa et al., 2021*). Here we demonstrate that both protein pairs contain a ToxA-like family member (Avr2, SIX8) and a structural family 4 member (SIX5, PSL1). Considering the predicted structural similarities, we were interested in testing whether *Fol* SIX8 and PSL1 interact.

We heterologously produced *Fol* SIX8[50-141] and PSL1[18-111] (*Figure 1—figure supplement 1E and F*) (after referred to as SIX8 and PSL1) and co-incubated the proteins before analysing by size-exclusion chromatography (SEC) (*Figure 4B*). The elution profile of PSL1 alone showed a major peak (~12.25 mL) at a volume consistent with a dimeric form of the protein, while SIX8 showed a major peak (~15 mL) consistent with a monomer (*Figure 4B*). Strikingly, when incubated together the major protein peaks migrate to ~12.8 mL. SDS-PAGE analysis confirmed the presence of PSL1 and SIX8, indicating that the migration of both proteins on SEC is altered after incubation (*Figure 4B*). These data are consistent with PSL1 and SIX8 forming a heterodimer.

To understand the structural basis of the interaction, we attempted to solve the structure of the complex, but we were unable to obtain crystals. We subsequently utilised AlphaFold2-Multimer (*Evans et al., 2021*) through ColabFold (*Mirdita et al., 2022*) to model the interaction. Manual inspection of the top 5 models (*Figure 4—figure supplement 1A*, top model shown in *Figure 4C*) demonstrated that the thiol side chain of a free cysteine in PSL1 (Cys 37) and SIX8 (Cys 58) co-localised in the dimer interface, suggesting that an inter-disulfide bond may mediate the interaction. To test this, we performed intact mass spectrometry of SIX8 and PSL1 (alone and post incubation) under non-reduced and reducing conditions. The mass observed from the incubated SIX8 and PSL1 non-reduced sample contained a predominant species consistent with the combined molecular weight of SIX8 and PSL1 (20,777 Da) (*Figure 4D*, *Figure 4—figure supplement 2G and H*). SIX8 and PSL1 failed to form a heterodimer with an unrelated protein containing a free cysteine, suggesting specificity in the

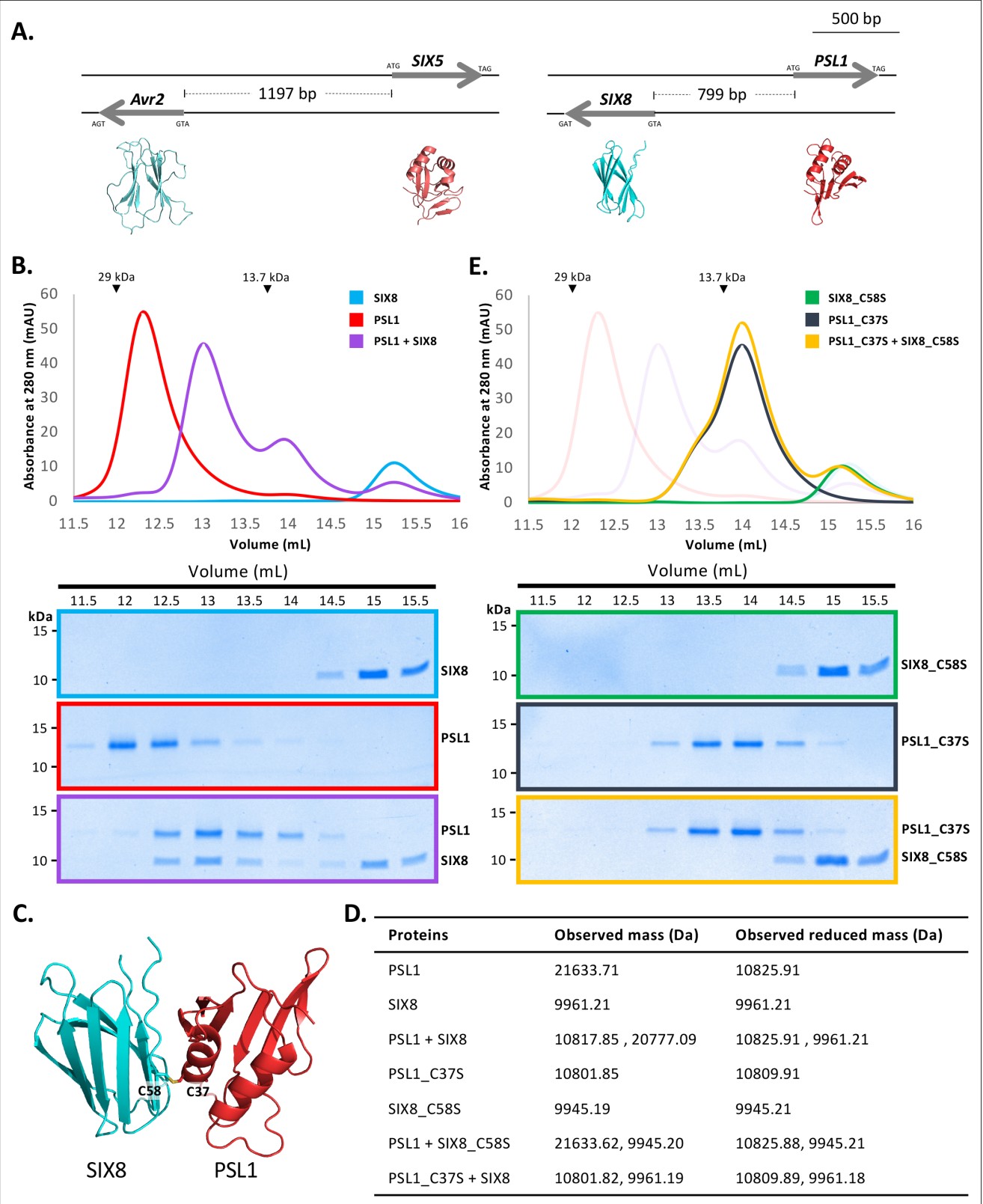

**Figure 4.** PSL1 and SIX8 interact in vitro mediated by an intermolecular disulfide bond. (**A**) Schematic representation of the *Avr2 (SIX3) – SIX5* and *SIX8 – PSL1* loci within *Fol*. AlphaFold2 models or experimentally solved protein structures are shown underneath. (**B**) Size-exclusion chromatograms of PSL1 alone (red), SIX8 alone (blue), PSL1 and SIX8 (purple) (following a 30 min incubation) separated over a Superdex S75 Increase SEC column (top panel). Equal concentrations of the protein were used (note the absorbance of SIX8 at 280 nm is ~0.3, resulting in a smaller absorbance and peak

*Figure 4 continued on next page*

*Figure 4 continued*

height). Indicated sizes above the chromatogram are based on protein standards run under similar conditions as presented in the manufacturer's column guidelines. Coomassie-stained SDS-PAGE gels depicting samples taken from 500 µL fractions corresponding to the volumes indicated above the gels, with molecular weights (left) and proteins (right) annotated (bottom panels). (**C**) Model of the SIX8-PSL1 complex generated by AlphaFold2-Multimer (top model shown). Co-localisation of Cys 58 from SIX8 and Cys 37 from PSL1 shown in stick form. (**D**) Observed masses of PSL1 and SIX8 protein mixtures by intact mass spectrometry (MS). Samples were treated with or without the reducing agent DTT prior to MS. The deconvoluted mass spectra of all proteins can be found in *Figure 4—figure supplements 2–4*. (**E**) As for (**B**) but with PSL1_C37S (black), SIX8_C58S (green), and PSL1_C37S and SIX8_C58S (yellow).

The online version of this article includes the following source data and figure supplement(s) for figure 4:

**Source data 1.** Unedited and uncropped SDS-PAGE gel for *Figure 4B*, SIX8 alone.

**Source data 2.** Unedited and uncropped SDS-PAGE gel for *Figure 4B*, SIX8 alone, with relevant bands labelled.

**Source data 3.** Unedited and uncropped SDS-PAGE gel for *Figure 4B*, PSL1 alone.

**Source data 4.** Unedited and uncropped SDS-PAGE gel for *Figure 4B*, PSL1 alone, with relevant bands labelled.

**Source data 5.** Unedited and uncropped SDS-PAGE gel for *Figure 4B*, PSL1+SIX8.

**Source data 6.** Unedited and uncropped SDS-PAGE gel for *Figure 4B*, PSL1+SIX8, with relevant bands labelled.

**Source data 7.** Unedited and uncropped SDS-PAGE gel for *Figure 4E*, SIX8_C58S alone.

**Source data 8.** Unedited and uncropped SDS-PAGE gel for *Figure 4E*, SIX8_C58S alone, with relevant bands labelled.

**Source data 9.** Unedited and uncropped SDS-PAGE gel for *Figure 4E*, PSL1_C37S alone.

**Source data 10.** Unedited and uncropped SDS-PAGE gel for *Figure 4E*, PSL1_C37S alone, with relevant bands labelled.

**Source data 11.** Unedited and uncropped SDS-PAGE gel for *Figure 4E*, PSL1_C37S+SIX8_C58S.

**Source data 12.** Unedited and uncropped SDS-PAGE gel for *Figure 4E*, PSL1_C37S+SIX8_C58S, with relevant bands labelled.

**Figure supplement 1.** Interaction between PSL1 and SIX8 mutants.

**Figure supplement 1—source data 1.** Unedited and uncropped SDS-PAGE gel for *Figure 4—figure supplement 1B*, PSL1_C37S+SIX8.

**Figure supplement 1—source data 2.** Unedited and uncropped SDS-PAGE gel for *Figure 4—figure supplement 1B*, PSL1_C37S+SIX8, with relevant bands labelled.

**Figure supplement 1—source data 3.** Unedited and uncropped SDS-PAGE gel for *Figure 4—figure supplement 1B*, PSL1+SIX8_C58S.

**Figure supplement 1—source data 4.** Unedited and uncropped SDS-PAGE gel for *Figure 4—figure supplement 1B*, PSL1+SIX8_C58S, with relevant bands labelled.

**Figure supplement 2.** Intact mass spectrometry analysis of the PSL1-SIX8 interaction.

**Figure supplement 3.** Intact mass spectrometry analysis of the PSL1-SIX8 interaction.

**Figure supplement 4.** Intact mass spectrometry analysis of the PSL1-SIX8 interaction.

**Figure supplement 5.** Amino acid sequence alignment of SIX12 against family 4 members reveals a similar cysteine spacing.

interaction (*Figure 4—figure supplements 2I and 3J–L*). Collectively, these data demonstrated that the SIX8-PSL1 heterodimer is mediated via a disulfide bond.

To confirm the involvement of the predicted residues involved, interaction with cysteine mutants of PSL1 and SIX8 (PSL1_C37S[18-111] and SIX8_C58S[50-141], hereafter referred to as PSL1_C37S and SIX8_C37S) were analysed (*Figure 4E*). When PSL1_C37S was incubated with SIX8_C37S or SIX8 alone, the heterodimer was not resolved via SEC (*Figure 4D*, *Figure 4—figure supplement 1B*). This was further confirmed using mass spectrometry (*Figure 4C*, *Figure 4—figure supplements 3Q and 4R–V*). We crystallised and solved the structure of SIX8_C58S at 1.28 Å (*Figure 1—figure supplement 1E*, *Figure 4—figure supplement 1C*) which confirms its inclusion within the ToxA-like structural family.

## The molecular basis of Avr1 recognition by the I receptor

The structural identification of the FOLD effector family provides an opportunity to understand their recognition by cognate immunity receptors. Here, we focussed on Avr1 (SIX4), which is recognised by the I immunity receptor leading to ETI and disease resistance (*Catanzariti et al., 2017*). Previous studies have shown co-expression of the *I* gene from the M82 tomato cultivar (I[M82]) with Avr1 in *Nicotiana benthamiana* leads to a cell death response, a proxy for ETI (*Catanzariti et al., 2017*). Conversely, co-expression with the allelic variant (i[Moneymaker]) from the susceptible cultivar Moneymaker does not lead to cell death as the receptor cannot recognise Avr1 (*Catanzariti et al., 2017*; *Figure 5B*). Here we sought to further define the recognition between Avr1 and I utilising the *N. benthamiana* system.

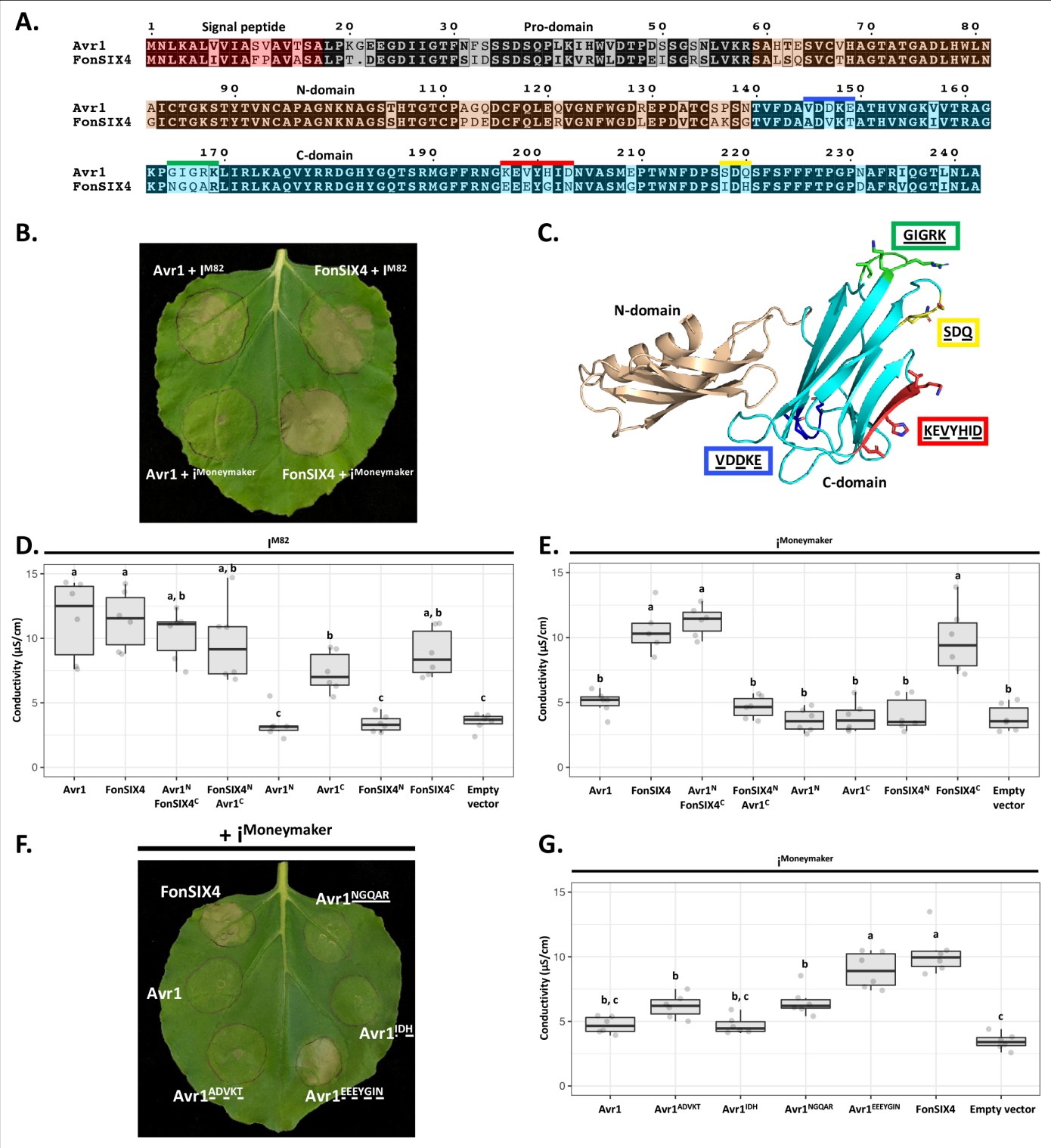

**Figure 5.** The C-domain of Avr1 mediates recognition by the I receptor. (**A**) Amino acid sequence alignment of Avr1 and FonSIX4, a homologue from *F. oxysporum* f. sp. *niveum*. The signal peptide, pro-domain, N-domain, and C-domain are highlighted in red, grey, beige, and blue, respectively. Within the C-domain, surface-exposed regions that differ between Avr1 and FonSIX4 are overlined. (**B**) Avr1 and FonSIX4 were transiently expressed in *Nicotiana benthamiana* with either I^M82 or i^Moneymaker via *Agrobacterium*-mediated transformation (n = 6). (**C**) The crystal structure of Avr1, showing the N- and C-domains in beige and light blue, respectively as represented in (**A**). Regions containing variant residues within the C-domain between Avr1 and FonSIX4 are coloured corresponding to the overlined colours in (**A**). Variant residues are underlined and represented in stick form. (**D**) Ion leakage conductivity of the Avr1 and FonSIX4 chimeric constructs, and N- and C-domains individually, when transiently co-expressed with I^M82 or (**E**) i^Moneymaker.

*Figure 5 continued on next page*

*Figure 5 continued*

Two additional independent experiments were repeated with similar results (*Figure 5—figure supplement 2*). (**F**) Leaf image and (**G**) ion leakage quantification of the Avr1 mutants (Avr1$\underline{A}^D\underline{V}^K\underline{T}$, Avr1$\underline{NGQAR}$, Avr1$\underline{I}^D\underline{H}$, Avr1$\underline{E}^E\underline{E}^Y\underline{G}^I\underline{N}$) when transiently co-expressed with i$^{Moneymaker}$ (n = 6). Variant residues between Avr1 and FonSIX4 are underlined. Six biological replicates for each construct were measured using an ion leakage assay. One-way ANOVA and post hoc Tukey's honestly significant difference tests were performed. Treatments that do not share a letter are significantly different from each other at p<0.05. Leaves were imaged 5 days post infiltration (dpi).

The online version of this article includes the following source data and figure supplement(s) for figure 5:

**Source data 1.** Conductivity measurements for *Figure 5D*.

**Source data 2.** Conductivity measurements for *Figure 5E*.

**Source data 3.** Conductivity measurements for *Figure 5G*.

**Figure supplement 1.** I receptor recognition of Avr1 and FonSIX4 mutants in N. benthamiana.

**Figure supplement 1—source data 1.** Unedited and uncropped blot for *Figure 5—figure supplement 1G*, western blots of Avr1 and FonSIX4 constructs with a C-terminal HA tag (Avr1, FonSIX4, Avr1$^N$ FonSIX4$^C$, FonSIX4$^N$ Avr1$^C$, Avr1$^{ADVKT}$, Avr1$^{NGQAR}$, Avr1$^{IDH}$, Avr1$^{EEEYGIN}$, and Empty vector).

**Figure supplement 1—source data 2.** Unedited and uncropped blot for *Figure 5—figure supplement 1G*, western blots of Avr1 and FonSIX4 constructs with a C-terminal HA tag (Avr1, FonSIX4, Avr1$^N$ FonSIX4$^C$, FonSIX4$^N$ Avr1$^C$, Avr1$^{ADVKT}$, Avr1$^{NGQAR}$, Avr1$^{IDH}$, Avr1$^{EEEYGIN}$, and Empty vector), with relevant bands labelled.

**Figure supplement 1—source data 3.** Unedited and uncropped blot for *Figure 5—figure supplement 1G*, Ponceau-stained membrane of Avr1 and FonSIX4 constructs with a C-terminal HA tag (Avr1, FonSIX4, Avr1$^N$ FonSIX4$^C$, FonSIX4$^N$ Avr1$^C$, Avr1$^{ADVKT}$, Avr1$^{NGQAR}$, Avr1$^{IDH}$, Avr1$^{EEEYGIN}$, and Empty vector).

**Figure supplement 1—source data 4.** Unedited and uncropped blot for *Figure 5—figure supplement 1G*, Ponceau-stained membrane of Avr1 and FonSIX4 constructs with a C-terminal HA tag (Avr1, FonSIX4, Avr1$^N$ FonSIX4$^C$, FonSIX4$^N$ Avr1$^C$, Avr1$^{ADVKT}$, Avr1$^{NGQAR}$, Avr1$^{IDH}$, Avr1$^{EEEYGIN}$, and Empty vector), with relevant bands labelled.

**Figure supplement 1—source data 5.** Unedited and uncropped blot for *Figure 5—figure supplement 1G*, western blots of Avr1 and FonSIX4 constructs with a C-terminal HA tag (Avr1, Avr1$^N$, Avr1$^C$, FonSIX4, FonSIX4$^N$, FonSIX4$^C$, and Empty vector).

**Figure supplement 1—source data 6.** Unedited and uncropped blot for *Figure 5—figure supplement 1G*, western blots of Avr1 and FonSIX4 constructs with a C-terminal HA tag (Avr1, Avr1$^N$, Avr1$^C$, FonSIX4, FonSIX4$^N$, FonSIX4$^C$, and Empty vector), with relevant bands labelled.

**Figure supplement 1—source data 7.** Unedited and uncropped blot for *Figure 5—figure supplement 1G*, Ponceau-stained membrane of Avr1 and FonSIX4 constructs with a C-terminal HA tag (Avr1, Avr1$^N$, Avr1$^C$, FonSIX4, FonSIX4$^N$, FonSIX4$^C$, and Empty vector).

**Figure supplement 1—source data 8.** Unedited and uncropped blot for *Figure 5—figure supplement 1G*, Ponceau-stained membrane of Avr1 and FonSIX4 constructs with a C-terminal HA tag (Avr1, Avr1$^N$, Avr1$^C$, FonSIX4, FonSIX4$^N$, FonSIX4$^C$, and Empty vector), with relevant bands labelled.

**Figure supplement 1—source data 9.** Unedited and uncropped blot for *Figure 5—figure supplement 1G*, western blots of Avr1 and FonSIX4 constructs with a C-terminal HA tag (Avr1, FonSIX4, FonSIX4$^{KEVYHID}$, and Empty vector).

**Figure supplement 1—source data 10.** Unedited and uncropped blot for *Figure 5—figure supplement 1G*, western blots of Avr1 and FonSIX4 constructs with a C-terminal HA tag (Avr1, FonSIX4, FonSIX4$^{KEVYHID}$, and Empty vector), with relevant bands labelled.

**Figure supplement 1—source data 11.** Unedited and uncropped blot for *Figure 5—figure supplement 1G*, Ponceau-stained membrane of Avr1 and FonSIX4 constructs with a C-terminal HA tag (Avr1, FonSIX4, FonSIX4$^{KEVYHID}$, and Empty vector).

**Figure supplement 1—source data 12.** Unedited and uncropped blot for *Figure 5—figure supplement 1G*, Ponceau-stained membrane of Avr1 and FonSIX4 constructs with a C-terminal HA tag (Avr1, FonSIX4, FonSIX4$^{KEVYHID}$, and Empty vector), with relevant bands labelled.

**Figure supplement 1—source data 13.** Conductivity measurements for *Figure 5—figure supplement 1D*.

**Figure supplement 2.** Ion leakage conductivity of different Avr1 and FonSIX4 chimeras expressed with I$^{M82}$ or i$^{Moneymaker}$.

**Figure supplement 2—source data 1.** Conductivity measurements for *Figure 5—figure supplement 2A*, left panel.

**Figure supplement 2—source data 2.** Conductivity measurements for *Figure 5—figure supplement 2A*, right panel.

**Figure supplement 2—source data 3.** Conductivity measurements for *Figure 5—figure supplement 2B*, left panel.

**Figure supplement 2—source data 4.** Conductivity measurements for *Figure 5—figure supplement 2B*, right panel.

To facilitate this, we identified homologues of Avr1 that possess natural residue variation. FonSIX4, a homologue of Avr1 from the watermelon pathogen, *F. oxysporum* f. sp. *niveum* (*Fon*), shares 79% identity with Avr1 (*Figure 5A*). Using the *N. benthamiana* assay, we show FonSIX4 is recognised by I receptors from both cultivars (I$^{M82}$ and i$^{Moneymaker}$) and cell death is dependent on the presence of I$^{M82}$ or i$^{Moneymaker}$ (*Figure 5B*, *Figure 5—figure supplement 1F*). FonSIX4 and Avr1 differ by 34 residues distributed across both N- and C-domains of the protein (*Figure 5A*). To narrow down the regions involved in recognition, we designed chimeric variants by swapping the N- and C-domains (Avr1$^N$-FonSIX4$^C$ and FonSIX4$^N$Avr1$^C$) (*Figure 5A and C*). When these were co-expressed with i$^{Moneymaker}$, the

cell death response, quantified using ion leakage assays (*Figure 5D and E*) and visual inspection (*Figure 5—figure supplement 1A*), suggests the C-domain of FonSIX4 is recognised by i$^{Moneymaker}$. We separated Avr1 and FonSIX4 proteins into their N- or C-domains and co-expressed with I$^{M82}$ or i$^{Moneymaker}$. Quantification using ion leakage assays demonstrate that the C-domains of Avr1 and FonSIX4 cause cell death when expressed with I$^{M82}$ and I$^{M82/Moneymaker}$, respectively. These data confirm the C-domain is sufficient for I receptor recognition (*Figure 5D and E*, *Figure 5—figure supplement 2*, see *Figure 5—figure supplement 1* for *N. benthamiana* leaf infiltration and protein accumulation data).

To understand how Avr1 can escape i$^{Moneymaker}$ recognition, we focussed on surface-exposed variant residues (underlined) mapping to four regions within the C-domain (*Figure 5A and C*). Four reciprocal swap mutants between Avr1 and FonSIX4 (Avr1$^{\underline{ADVKT}}$, Avr1$^{\underline{IDH}}$, Avr1$^{\underline{NGQAR}}$, Avr1$^{\underline{EEEYGIN}}$) were co-expressed with i$^{Moneymaker}$ to identify the residues required for FonSIX4 recognition. Avr1$^{\underline{EEEYGIN}}$ showed consistent ion leakage and cell death similar to FonSIX4 (*Figure 5F and G*), whereas ion leakage quantification for the other three mutants (Avr1$^{\underline{ADVKT}}$, Avr1$^{\underline{IDH}}$, Avr1$^{\underline{NGQAR}}$) was statistically similar to the non-recognised Avr1 (*Figure 5G*). The reciprocal mutations in FonSIX4 (FonSIX4$^{\underline{KELVYHID}}$) significantly reduced ion leakage and cell death response when co-expressed with i$^{Moneymaker}$ compared to FonSIX4 (*Figure 5—figure supplement 1D and E*, see *Figure 5—figure supplement 1G* for protein accumulation data). Collectively, these data show that the C-domain in Avr1 is recognised by I$^{M82}$, and surface-exposed residues in the C-domain allow Avr1 to escape recognition by i$^{Moneymaker}$.

## Discussion

Pathogenic fungi are in a continuous arms race with their plant hosts. To aid virulence, but avoid detection, effectors evolve rapidly, causing significant diversity at the amino acid sequence level (*Stergiopoulos and de Wit, 2009*). An emerging theme in fungal effector biology is the classification of effectors into families based on structural similarity (*Outram et al., 2022*). Here, we demonstrate that despite their sequence diversity, the *Fol* SIX effectors can be classified into a reduced set of structural families. This observation has implications for functional studies of SIX effectors, and ultimately our understanding of the infection strategies used by *F. oxysporum*.

### Expanding the structural classes in fungal effectors

To date, five fungal effector families have been defined based on experimentally determined structural homology, including the MAX (*de Guillen et al., 2015*), RALPH (*Spanu, 2017*; *Pennington et al., 2019*; *Pedersen et al., 2012*), ToxA-like (*Di et al., 2017*; *Sarma et al., 2005*; *Wang et al., 2007*), LARS (*Lazar et al., 2022*; *Blondeau et al., 2015*), and FOLD effectors, defined here. Effectors that fall within many of these structural families are shared across distantly related fungal species. The ToxA-like family includes effectors from fungi that group to both divisions of higher-fungi (Basidiomycota and Ascomycota fungi) (*Di et al., 2017*; *Sarma et al., 2005*; *Wang et al., 2007*). The MAX effector family were originally defined as AVR effectors from *M. oryzae* and ToxB from *P. tritici-repentis* (*de Guillen et al., 2015*) but pattern-based sequence searches suggest they are widely distributed amongst the Dothideomycetes and Sordariomycetes (*de Guillen et al., 2015*; *Petit-Houdenot et al., 2020*). Similarly, LARS effectors, defined in *Leptosphaeria maculans* and *Fulvia fulva,* have structural homologues predicted in at least 13 different fungal species (*Lazar et al., 2022*). Based on sequence homologues alone, FOLD effectors are well dispersed in fungi with homologues amongst the Sordariomycetes including many *formae speciales* of *F. oxysporum, Colletotrichum,* and *Ustilaginoidea.* Based on structural comparison of the AlphaFold2 structural database, we show that it is extended to fungi in three divisions, including plant pathogens and symbionts. This was supported by a recent study modelling the secretomes of arbuscular mycorrhizal fungi which found enlarged and diversified gene families encoded proteins predicted to share the FOLD effector structure (*Teulet et al., 2023*). The exclusive presence of FOLD effectors in plant-colonising fungi may suggest that they facilitate plant colonisation in pathogenic and symbiotic fungi (*Teulet et al., 2023*).

### Effector structure prediction

Experimentally determining the structures of fungal effectors is not a trivial undertaking. From challenges associated with effector protein production through to hurdles related to structure solution (such as experimental phasing), the research time required to determine an effector structure

experimentally ranges from months to many years (sometimes never). Not surprisingly, any reliable structural modelling methods are welcomed by researchers interested in effector biology. To this end, several recent studies have used effector structure prediction to expand our understanding of plant–microbe interactions (*Bauer et al., 2021*; *Seong and Krasileva, 2021*).

Work by Bauer and colleagues, prior to the release of AlphaFold2, used structural modelling to show that numerous recognised Avr effectors from the barley powdery mildew-causing fungal pathogen *Blumeria graminis* (*Bgh*) are members of the RALPH effectors class (*Bauer et al., 2021*). Seong and Krasileva used similar structural modelling approaches to predict the folds of ~70% of the *Magnaporthe oryzae* secretome (*Seong and Krasileva, 2021*). In doing so, they suggested an expansion in the number of MAX effectors and identified numerous sequence-unrelated groups of structural homologues (putative structural classes) within *M. oryzae*. Making use of AlphaFold2, Yan and colleagues show that structurally conserved effectors, including the MAX effector family, from *M. oryzae* are temporally co-expressed during the infection process (*Yan et al., 2023*). In the largest comparison study to date, Seong and Krasileva carried out a large comparative structural genomics study of fungal effectors utilising AlphaFold2 (*Seong and Krasileva, 2023*). Their findings support the hypothesis that the structurally conserved effector families are the result of divergent evolution and support previous finding that the structural landscape of effectors is more limited than what is suggested by sequence diversification.

Here, we were in a unique position to apply and benchmark AlphaFold2 against experimentally determined structures for *Fol* effector prediction. We subsequently used AlphaFold2 to demonstrate that, within the repertoire of effectors we tested, up to five sequence-unrelated structural families are secreted during *Fol* infection. There are numerous caveats in relying solely on AlphaFold2 to generate structural models of effectors. The accuracy of models generated by AlphaFold2 can decline in cases with low numbers of homologues (~30 sequences in the multiple sequence alignment [MSA]) (*Jumper et al., 2021*). This may help explain the low-confidence prediction for SIX4 (Avr1) (*Figure 2—figure supplement 1A*), which is only distributed in a few ff. spp. of *F. oxysporum*. This poses a potential issue for predicting the structures of fungal effectors that lack homologues. In our hands, we have had mixed results when comparing several unpublished effector structures experimentally determined in our lab to AlphaFold2 models. In some instances, the models are wrong, for example, AvrSr50 (*Ortiz et al., 2022*); however, in these cases the AlphaFold2 predictions reported low-confidence scores, an important criterion for the assessment of model reliability. Despite this, AlphaFold2 models were critical in solving the structure of SIX6 and SIX8 as templates for molecular replacement. This negated the need to derivatise our crystals, a process that we had struggled with for SIX6 crystals, significantly reducing the time and research effort to determine the experimental structures.

## Structural classes: A starting point for functional characterisation

Given their lack of sequence identity to proteins of known function or conserved motifs, structural determination of effectors is often pursued to provide functional insight and understanding of residues involved in recognition. The existence of structural families of effectors raises the question of whether links can now be made concerning their function based on structural similarities. Unfortunately, the FOLD effectors share little overall structural similarity with known structures in the PDB. However, at a domain level, the N-domain of FOLD effectors have structural similarities with cystatin cysteine protease inhibitors (PDB code: 4N6V; PDB code: 5ZC1) (*Park et al., 2018*; *Renko et al., 2014*), while the C-domains have structural similarities with tumour necrosis factors (PDB code: 6X83) (*Dietrich et al., 2021*) and carbohydrate-binding lectins (PDB code: 2WQ4) (*Sulák et al., 2010*). Though a functional link has not yet been established, the information gleaned from the FOLD effector structures gives us a starting point for further functional characterisation, with various avenues now being explored.

Interestingly, the predicted models for SIX9 and SIX11 within family 3 have structural homology with RNA-binding proteins (PDB code: 3NS6; PDB code: 5X3Z) (*Iwaoka et al., 2017*; *Khoshnevis et al., 2010*), unrelated to RALPH effectors. Despite this structural homology, close inspection of these models suggests that RNA binding is unlikely as in both models the putative RNA-binding surface is disrupted by a disulfide bond.

The putative family 4 effectors (SIX5, SIX14, PSL1, and PSL2) have structural homology with KP6 effectors and heavy metal-associated (HMA) domains. Metal binding within HMA domains is facilitated

by conserved cysteine residues (*Bull and Cox, 1994*); however, their absence in the family 4 effectors suggests they are unlikely to have this activity.

The putative family 5 effectors (FOXGR_015522 and FOXG_18699) have structural homology with different proteins within the PDB. FOXGR_015522 is structurally similar to plant defensins (PDB code: 6MRY; PDB code: 7JN6) (*Bleackley et al., 2020*; *Lay et al., 2019*) and $K^+$ channel-blocking scorpion toxins (PDB code: 1J5J; PDB code: 2AXK) (*Prochnicka-Chalufour et al., 2006*; *Korolkova et al., 2002*). FOXG_18699 has structural homology with the C-terminal domain of bacterial arginine repressors (PDB code: 1XXB; PDB code: 3CAG) (*Van Duyne et al., 1996*; *Cherney et al., 2008*).

## A structural explanation for functional effector pairs

One interesting outcome of this study is a link between structural families and cooperative interactions between effectors. The ToxA-like effectors, Avr2 and SIX8, are known to form functional effector pairs with SIX5 and PSE1 (PSL1-homologue), respectively (*Ma et al., 2015*; *Ayukawa et al., 2021*). According to our modelling work, both SIX5 and PSL1 are members of structural family 4. *Avr2* and *SIX5* are adjacent divergently transcribed genes on *Fol* chromosome 14 and the protein products have been shown to physically interact (*Ma et al., 2015*). Likewise, *SIX8* and *PSL1* are adjacent divergently transcribed genes in the *Fol* genome, and we demonstrate here a physical interaction between the proteins. The AlphaFold2-Multimer models of the SIX8 and PSL1 heterodimer drew our attention to the inter-disulfide bond between SIX8 and PSL1 required for the interaction, which we confirmed experimentally. While these residues are conserved in *Focn* SIX8 and PSE1, the Avr2 structure and SIX5 model lack free cysteine residues, suggesting a different mode of interaction.

Interestingly, two other *SIX* genes also form a divergently transcribed gene pair on *Fol* chromosome 14. *SIX7* (ToxA-like family) and *SIX12* possess start codons 2319 base-pairs apart and potentially share a common promoter. While SIX12 did not group with any structural families, the AlphaFold2 model had a very low prediction confidence (35.5%). On closer inspection of the sequence, we observed that the cysteine spacing in SIX12 closely resembles other family 4 members (*Figure 4—figure supplement 5*), which suggests that SIX12 may also be a family 4 member. We therefore speculate that SIX7 and SIX12 may function together, as described for the Avr2/SIX5 and SIX8/PSL1 pairs.

## Are experimentally derived effector structures still worth the effort?

The potential of machine-learning structural-prediction programs, such as AlphaFold2, heralds an exciting era, especially for a field that has long suffered from a lack of prediction power based on effector sequences. A question now emerges; when prediction model confidence is high, should we bother solving structures experimentally? The answer such a question will always depend on what the structure is being used for. Ultimately, structural models, whether experimentally or computationally derived, represent information to base and/or develop a hypothesis to subsequently test. Here, we demonstrate the power of structure prediction in combination with experimentation, both for validating models and understanding protein:protein interaction interfaces. One interesting observation we made was that while the AphaFold2-Multimer models of the SIX8 and PSL1 heterodimer were sufficient to highlight the cysteine residues required for mediating the interaction, the models and interaction interfaces differed significantly (*Figure 4—figure supplement 1A*). When the modelling was repeated with the SIX8$^{C58S}$ experimentally derived structure included as a template, the interaction models and heterodimer interface were of higher quality and essentially identical (*Figure 4—figure supplement 1E*). This observation can be retrospectively reconciled. The region of SIX8 involved in the interaction with PSL1 was modelled incorrectly by AlphaFold2 when compared to the structure (*Figure 4—figure supplement 1D*). Collectively, these data highlight that some models are good enough, but others maybe better.

## Effector structural classes and understanding receptor recognition

Understanding the structural basis of plant immunity receptor–effector interactions represents a key step towards engineering plant immunity receptors with novel specificities. Recent structures of nucleotide-binding domain leucine-rich repeat (NLR) proteins reveal exquisite details of these direct interactions (*Förderer et al., 2022*; *Ma et al., 2020*; *Martin et al., 2020*). The FOLD effectors, Avr1 and Avr3, are recognised by different classes of immunity receptors; I, an LRR-RP (*Catanzariti et al., 2017*), and I-3, a SRLK (*Catanzariti et al., 2015*). While the mode of recognition has not yet been

described for Avr3, we demonstrate here that Avr1 is recognised at the C-domain (*Figure 5*). This is significant because it demonstrates that different immunity receptor classes can recognise structural homologues. It might also help explain the function of Avr1 during *Fol* infection. When Houterman and colleagues identified Avr1, they demonstrated that it could suppress plant immunity conferred by the *I-2* and *I-3* receptors (*Houterman et al., 2008*). Considering our structural understanding of these FOLD effectors, it is plausible that Avr1 achieves suppression of I-3-mediated immunity by preventing Avr3 recognition through competitive inhibition. The LARS effectors represent another example of effectors that can activate and suppress resistance-gene-mediated immunity. AvrLm4-7 can prevent recognition of AvrLm3 and AvrLm9 (all LARS structural homologues; *Lazar et al., 2022*), by their cognate Rlm receptors (*Plissonneau et al., 2016*; *Ghanbarnia et al., 2018*). *Rlm4*, *Rlm7*, and *Rlm9* all encode for wall-associated kinases (*Haddadi et al., 2022*; *Larkan et al., 2020*), but the identity of *Rlm3* remains unknown. These studies demonstrate that members of at least two different structural effector families can suppress immunity triggered by structurally homologous effectors.

Collectively, the results presented here will aid future studies in understanding the molecular basis of *F. oxysporum* effector function and recognition, and by extension, the design and engineering of immunity receptors with novel recognition specificities to help protect plants against *Fusarium* wilt disease.

## Materials and methods

### Vectors and gene constructs

SIX6, Avr1Thrombin, SIX6-TEV, SIX8Thrombin, SIX8_C58SThrombin, PSL1, PSL1_C37S, and SIX13 coding sequences (without their signal peptides as determined by SignalP-5.0) were codon optimised for expression in *Escherichia coli* and synthesised with Golden-Gate-compatible overhangs by Integrated DNA Technologies (IDT, Coralville, USA) (*Supplementary file 4*). The Kex2 cleavage motif of Avr1 and SIX8 were replaced with a thrombin cleavage motif, and TEV protease cleavage motif for SIX6 for pro-domain processing. Avr1 and Avr3 coding sequences were PCR amplified using *Fol* cDNA as a template with primers containing Golden-Gate-compatible overhangs. All genes for *E. coli* expression were cloned into a modified, Golden-Gate-compatible, pOPIN expression vector (*Bentham et al., 2021*). The final expression constructs contained N-terminal 6xHis-GB1-tags followed by 3C protease recognition sites. The Golden-Gate digestion, ligation reactions, and PCR were carried out as described by *Iverson et al., 2016*. Avr1 and FonSIX4 mutant sequences without the signal peptide were synthesised with compatible overhangs by IDT (*Supplementary file 4*) and cloned into the pSL vector containing the *Nicotiana tabacum* PR1 signal peptide using the In-fusion cloning kit (Takara Bio USA Inc, San Jose, USA) to allow efficient secretion in *N. benthamiana* via *Agrobacterium*-mediated expression. For tagged constructs, Avr1 and FonSIX4 mutant sequences and 3xHA tag were amplified with PCR and assembled using In-fusion cloning into the pSL vector. All of the primers were synthesised by IDT (*Supplementary file 5*). All constructs were verified by sequencing.

### Protein expression and purification

Sequence-verified constructs were co-expressed with CyDisCo in SHuffle T7 Express C3029 (New England Biolabs, Ipswich, USA) and purified as previously described (*Yu et al., 2021*). For Avr3, the buffers used after fusion tag cleavage were altered slightly to increase protein stability and a second IMAC step was excluded after the cleavage of the N-terminal fusion tag. During the cleavage step, the protein was dialysed into a buffer containing 10 mM MES pH 5.5 and 300 mM NaCl. The SEC HiLoad 16/600 Superdex 75 pg column (Cytiva) was equilibrated with a buffer containing 10 mM MES pH 5.5 and 150 mM NaCl.

For biochemical and crystallisation studies, Avr1 and SIX8 with an internal thrombin cleavage site, and SIX6 with an internal TEV protease cleavage site for pro-domain removal were processed with 2–4 units of thrombin from bovine plasma (600–2000 NIH units/mg protein) (Sigma-Aldrich Inc, St. Louis, USA) per mg of protein at 4°C or TEV protease (produced in-house) until fully cleaved. Fully cleaved proteins were purified further by SEC using a HiLoad 16/600 or HiLoad 26/600 Superdex 75 pg column (Cytiva) equilibrated with a buffer containing 10 mM HEPES pH 7.5 or 8.0 and 150 mM NaCl. Proteins were concentrated using a 10 or 3 kDa molecular weight cutoff Amicon centrifugal

concentrator (MilliporeSigma, Burlington, USA), snap-frozen in liquid nitrogen, and stored at –80°C for future use.

## Intact mass spectrometry

Proteins were adjusted to a final concentration of 6 µM in 0.1% (v/v) formic acid (FA) for HPLC-MS analysis for untreated samples. For reduced samples, DTT was added to the protein to a final concentration of 10 mM. Proteins were incubated at 60°C for 30 min and adjusted to 6 µM in 0.1% (v/v) FA. Intact mass spectrometry on all proteins was carried out as described previously (*Yu et al., 2021*). Data were analysed using the Free Style v.1.4 (Thermo Fisher Scientific) protein reconstruct tool across a mass range of m/z 500–2000 and compared against the theoretical (sequence-based) monoisotopic mass.

## Circular dichroism (CD) spectroscopy

The CD spectra of purified effectors of interest were recorded on a Chirascan spectrometer (Applied Photophysics Ltd, UK) at 20°C. Samples were diluted to 10 µM in a 20 mM sodium phosphate buffer at pH 8.0. Measurements were taken at 1 nm wavelength increments from 190 nm to 260 nm. A cell with a pathlength of 1 mm, a bandwidth of 0.5 nm, and response time of 4 s were used, with three accumulations. The data were averaged and corrected for buffer baseline contribution, and visualised using the webserver CAPITO tool with data smoothing (*Wiedemann et al., 2013*). CD analysis was performed on all purified proteins (*Figure 1—figure supplement 3*).

## Crystallisation, diffraction data collection, and crystal structure determination

Initial screening to determine crystallisation conditions was performed at a concentration of 9.5 mg/mL for Avr3[22-284], 10 mg/mL for Avr1[18-242], Avr1[59-242], SIX8[50-141], and PSL1[18-111], 15 mg/mL for SIX6[17-225] and SIX6[58-225], 25 mg/mL for SIX8_C58S[19-141], 18 mg/mL for SIX8_C58S[50-141] and PSL1_C37S[18-111], and 14 mg/mL for SIX8-PSL1 complex and SIX13 with and without Kex2 protease in 96-well MRC 2 plates (Hampton Research) at 18°C using the sitting-drop vapour-diffusion method and commercially available sparse matrix screens. For screening, 150 nL protein solution and 150 nL reservoir solution were prepared on a sitting-drop well using an NT8-Drop Setter robot (Formulatrix, USA). The drops were monitored and imaged using the Rock Imager system (Formulatrix) over the course of a month.

For Avr1[18-242], SIX6[17-225], SIX8[50-141], PSL1[18-111], SIX8-PSL1 complex, and SIX13[22-293], no crystals were obtained from the different sparse matrix screens trialled. From initial screening, crystals with the best morphology for Avr3[22-284] were obtained in (1) 0.2 M lithium sulfate, 0.1 M Bis-Tris pH 6.5, and 25% (w/v) PEG 3350 (SG1 screen: condition D10), and (2) 0.2 M ammonium sulfate, 0.1 M Bis-Tris pH 6.5, and 25% (w/v) PEG 3350 (SG1 screen: condition F5). Crystals were visible after a period of 3 d and continued to grow for 3 wk after initial setup. Replicate drops with 1 µL protein solution at 9.5 mg/mL and 1 µL reservoir solution were setup in 24-well hanging-drop vapour-diffusion plates and produced crystals within 4 d that continued to grow over 1 mo. No crystal optimisation was needed for Avr3, with the final conditions being (1) 0.2 M ammonium sulfate, 0.1 M Bis-Tris pH 6.5, and 25% (w/v) PEG 3350, and (2) 0.2 M lithium sulfate, 0.1 M Bis-Tris pH 6.5, and 25% (w/v) PEG 3350. For Avr1[59-242], crystals with the best morphology were obtained in (1) 0.2 M ammonium sulfate, 0.1 M sodium acetate pH 4.6, and 25% (w/v) PEG 4000 (SG1 screen: condition C1), and (2) 0.2 M ammonium sulfate, 30% (w/v) PEG 8000 (SG1 screen: condition D7) within 1 d of initial setup. Crystal optimisation was carried out in 24-well hanging-drop vapour-diffusion plates at 18°C. The final optimised condition for Avr1[59-242] was 0.2 M ammonium sulfate, 0.1 M sodium acetate pH 4.5, 17.5% (w/v) PEG 4000 at a protein concentration of 7 mg/mL with microseeding over a period of 3 wk. For SIX6[58-225], crystals were obtained in 0.2 M ammonium tartrate and 20% (w/v) PEG 3350 (SG1 screen: condition G9) 40 d after initial setup. Crystals were picked directly from the sparse matrix screen. For SIX8_C58S[50-141], crystals were obtained in 0.17 M ammonium sulfate, 15% (v/v) glycerol, and 25.5% (w/v) PEG 4000 (JCSG screen: condition D9) a week after initial setup. Crystals were picked directly from the sparse matrix screen. For SIX13, Kex2 protease was added to the protein at a 1:200 protease to protein ratio prior to crystal tray setup. Crystals with the best morphology were obtained in (1) 0.2 M lithium sulfate, 0.1 M Bis-Tris pH 6.5, and 25% (w/v) PEG 3350 (SG1 screen: condition D10), and (2) 0.2 M ammonium sulfate, 0.1 M Bis-Tris pH 6.5, and 25% (w/v) PEG 3350 (SG1 screen: condition F5) within 2 d of initial setup.

Crystals were optimised using hanging-drop vapour-diffusion plates and the final optimised condition for SIX13 was 0.2 M lithium sulfate, 0.1 M Bis-Tris pH 6.5, and 25% (w/v) PEG 3350 at a protein concentration of 14 mg/mL. For PSL1_C37S[18-111], crystals were obtained in 70% (v/v) MPD and 0.1 M HEPES pH 7.5 within 3 d after initial setup. Crystal optimisation was carried out in 24-well hanging-drop vapour-diffusion plates at 18°C. The final optimised condition for PSL1_C37S[18-111] was 62% (v/v) MPD and 0.1 M HEPES pH 7.5 at a protein concentration of 17.5 mg/mL.

Crystals were transferred into a cryoprotectant solution containing reservoir solution and 15% (v/v) ethylene glycol, 20% (v/v) glycerol, or 10% (v/v) ethylene glycol and 10% (v/v) glycerol. No cryoprotecting was required for SIX8_C58S[50-141] and PSL1_C37S[18-111] crystals as the conditions contained sufficient cryoprotectant (glycerol and MPD, respectively) within the crystallisation condition. For experimental phasing, Avr3[22-284] and Avr1[59-242] crystals were soaked in a cryoprotectant solution containing 0.5 M or 1 M sodium bromide and vitrified in liquid nitrogen. The datasets for bromide-soaked crystals were collected on the MX1 beamline at the Australian Synchrotron (*Cowieson et al., 2015*; *Supplementary file 1*). The datasets were processed in XDS (*Kabsch, 2010*) and scaled with Aimless in the CCP4 suite (*Evans and Murshudov, 2013*; *Winn et al., 2011*). The CRANK2 pipeline in CCP4 was used for bromide-based SAD phasing (*Skubák and Pannu, 2013*; *Skubák et al., 2018*). Models were refined using phenix.refine in the PHENIX package (*Afonine et al., 2012*) and model building between refinement rounds was done in COOT (*Emsley et al., 2010*). The models were used as a template for molecular replacement against high-resolution native datasets collected on the MX2 beamline at the Australian Synchrotron (*Aragão et al., 2018*). Automatic model building was done using AutoBuild (*Terwilliger et al., 2008*), and subsequent models were refined with phenix.refine and COOT. For SIX6[58-225] and SIX8_C58S[50-141], high-confidence ab initio models were generated with AlphaFold2 (*Figure 2—figure supplement 1*), which was used as a template for molecular replacement against a native dataset collected on the MX2 beamline at the Australian Synchrotron. The resultant structure was refined as described above.

## Structural modelling and structural alignment

Structural models were generated with Google DeepMind's AlphaFold2 using the amino acid sequences of SIX effectors and candidates without the signal peptide, as predicted by SignalP-5.0 (*Almagro Armenteros et al., 2019*) and predicted pro-domain by searching for a Kex2 cleavage motif (KR, RR or LxxR) if present (*Outram et al., 2021b*; *Supplementary file 3*; *Figure 3—figure supplement 1*). For AlphaFold2 predictions, the full databases were used for MSA construction. All templates downloaded on 20 July 20 2021 were allowed for structural modelling. For each of the proteins, we produced five models and selected the best model (ranked_0.pdb) (AlphaFold2-generated files are shown in *Source data 1*). Pairwise alignments of the structural models generated by AlphaFold2 and the experimentally determined structures of Avr1 (PDB code: 7T6A), Avr3 (PDB code: 7T69), SIX6 (PDB code: 8EBB), and SIX8 (PDB code: 8EB9) were generated using the DALI server all against all function (*Holm, 2022*). Structural similarity between the pairwise alignments were measured using Z-scores from the DALI server.

## Distribution of FOLD family members across fungi

Structure-based searches to determine the distribution of FOLD effectors across other phytopathogens were carried out by searching the experimentally determined Avr1, Avr3, and SIX6 structures against available structure databases (Uniprot50, Proteome, Swiss-Prot) using the Foldseek webserver (*van Kempen et al., 2023*) using a 3Di search limited to fungi. An e-value cutoff of 0.01 was used, and non-plant-associated fungi were removed as well as duplicated results for final analysis. Proteins below 100 amino acids and above 500 amino acids were filtered, out and remaining structural hits were manually inspected for similarity to FOLD effectors.

## Interaction studies between PSL1 and SIX8

To investigate the PSL1 and SIX8 interaction in vitro, ~140 μg of PSL1[18-111] and SIX8[50-141] individually and ~140 μg PSL1[18-111] and 140 μg of SIX8[50-141] together were injected onto a Superdex 75 Increase 10/300 (Cytiva) column pre-equilibrated in 20 mM HEPES pH 7.5, 150 mM NaCl, after a 30 min room temperature incubation. To investigate the residues responsible for the interaction, SIX8_C58S[50-141] and PSL1_C37S[18-111] mutants were used instead. Samples across the peaks were then analysed by

Coomassie-stained SDS-PAGE. To investigate the mode of interaction, PSL1 and SIX8 proteins and mutants at 10 μM were incubated individually or together for 1 hr at room temperature. An unrelated protein with a free cysteine (AvrSr50[RKQQC]) (*Ortiz et al., 2022*) was used to assess the specificity of the PSL1-SIX8 interaction. Proteins were analysed by intact mass spectrometry with or without the addition of DTT as described above.

### *Agrobacterium*-mediated gene expression in *N. benthamiana*

*Agrobacterium tumefaciens* (GV3101) cultures containing the pSL constructs and pSOUP (*Hellens et al., 2000*) were diluted to an $OD_{600}$ of 1.0 in 10 mM MES pH 5.5 buffer containing 10 mM $MgCl_2$ and 200 μM acetosyringone and incubated in the dark for 2 hr. For co-infiltrations, cultures were mixed together in equal volumes. Resuspensions were infiltrated into 4–5-week-old *N. benthamiana* leaves. Infiltrated plants were kept in a 25°C controlled temperature room with a 16 hr photoperiod. Leaves were imaged 4–7 days post infiltration dpi.

### Ion leakage assay

Six biological replicates each consisting of three leaf discs (7 mm diameter) were harvested from leaves infiltrated with *Agrobacterium* 20–24 hr post infiltration and incubated in 7 mL of water in a 6-well culture plate. The water was replaced after 40–60 min. The leaf discs were incubated in water at room temperature, and the conductivity was measured after 24–48 hr.

### Immunoblot analysis of proteins expressed in *N. benthamiana*

*N. benthamiana* leaves infiltrated with *A. tumefaciens* cultures were harvested 3 dpi. Leaf tissue was frozen in liquid nitrogen, ground into a powder, and resuspended in 3× Laemmli buffer containing 0.2 mM DTT and 5 M urea to extract proteins. Samples were boiled for 10 min and centrifuged at $13,000 \times g$ to remove leaf debris. Proteins were separated by SDS-PAGE and transferred by electroblotting onto PVDF membranes. Protein blots were probed with anti-HA antibodies conjugated to horseradish peroxidase (Roche, Switzerland, 12013819001, 1:4000). Immunoblots were visualised with Pierce ECL Plus Western Blotting Substrate (Thermo Fisher Scientific) as described by the manufacturer. Membranes were stained with Ponceau S to assess protein loading.

### Materials availability statement

All newly created materials generated in this study are freely available from the corresponding authors without restriction (shipping costs may apply).

## Acknowledgements

This work was supported by the Australian Research Council (ARC DP200100388 DJ/SW) and the Australian Academy of Science (Thomas Davies Grant). SW was funded by an ARC Future Fellowship (FT200100135) and supported by the ANU Future Scheme (35665). LM was funded by an ARC Discovery Early Career Researcher Award (DE170101165). AS was a recipient of the AINSE Honours Scholarship Program, and DY and CM held an AINSE Postgraduate Research Award. PK was supported by a Netaji Subhas ICAR International Fellowship. The authors acknowledge the use of the ANU crystallisation facility. This research was undertaken in part using the MX2 beamline at the Australian Synchrotron, part of ANSTO, and made use of the Australian Cancer Research Foundation (ACRF) detector. The authors acknowledge use of the Australian Synchrotron MX facility and thank the staff for their support. The coordinates and structure factors for Avr1, Avr3, SIX6, and SIX8 have been deposited in the Protein Data Bank with accession numbers 7T6A, 7T69, 8EBB, and 8EB9, respectively.

## Additional information

### Funding

| Funder | Grant reference number | Author |
|---|---|---|
| Australian Research Council | DP200100388 | David A Jones |
| Australian Research Council | FT200100135 | Simon J Williams |
| Australian Research Council | DE170101165 | Lisong Ma |
| Australian Institute of Nuclear Science and Engineering | Postgraduate Research Award | Daniel S Yu Carl L McCombe |
| Australian National University | Future scheme | Simon J Williams |
| Australian Academy of Science | Thomas Davies Grant | Simon J Williams |
| ICAR - National Agricultural Science Fund | PhD International Fellowship | Pravin B Khambalkar |

The funders had no role in study design, data collection and interpretation, or the decision to submit the work for publication.

### Author contributions

Daniel S Yu, Conceptualization, Data curation, Formal analysis, Validation, Investigation, Visualization, Methodology, Writing – original draft, Writing – review and editing; Megan A Outram, Conceptualization, Formal analysis, Investigation, Methodology, Writing – original draft, Writing – review and editing; Ashley Smith, Carl L McCombe, Pravin B Khambalkar, Sharmin A Rima, Xizhe Sun, Lisong Ma, Investigation, Writing – review and editing; Daniel J Ericsson, Data curation, Formal analysis, Writing – review and editing; David A Jones, Resources, Supervision, Funding acquisition, Writing – review and editing; Simon J Williams, Conceptualization, Resources, Data curation, Formal analysis, Supervision, Funding acquisition, Validation, Methodology, Writing – original draft, Project administration, Writing – review and editing

### Author ORCIDs

Daniel S Yu http://orcid.org/0000-0003-0454-7989
Megan A Outram https://orcid.org/0000-0003-4510-3575
Carl L McCombe http://orcid.org/0000-0001-9347-8879
Lisong Ma http://orcid.org/0000-0001-9288-869X
Daniel J Ericsson http://orcid.org/0000-0001-5101-9244
Simon J Williams https://orcid.org/0000-0003-4781-6261

Reviewer #1 (Public Review): https://doi.org/10.7554/eLife.89280.3.sa1
Reviewer #2 (Public Review): https://doi.org/10.7554/eLife.89280.3.sa2
Reviewer #3 (Public Review): https://doi.org/10.7554/eLife.89280.3.sa3
Author Response https://doi.org/10.7554/eLife.89280.3.sa4

## Additional files

### Supplementary files

• Supplementary file 1. X-ray data collection, structure solution, and refinement statistics for Avr1, Avr3, SIX6, and SIX8.
• Supplementary file 2. Putative fungal FOLD effectors identified using the Foldseek webserver.
• Supplementary file 3. Amino acid sequence inputs for AlphaFold2.
• Supplementary file 4. DNA sequences of synthesised gene fragments used in this study.

- Supplementary file 5. Primers used in this study.
- Source data 1. AlphaFold2 pdb files generated.
- MDAR checklist

## Data availability

All data generated or analysed during this study are included in the manuscript and supporting files. The coordinates and structure factors for Avr1, Avr3, SIX6 and SIX8 have been deposited in the Protein Data Bank with accession number 7T6A, 7T69, 8EBB and 8EB9, respectively.

The following datasets were generated:

| Author(s) | Year | Dataset title | Dataset URL | Database and Identifier |
|---|---|---|---|---|
| Yu S, Outram MA, Ericsson DJ, Jones DA, Williams SJ | 2023 | Crystal structure of Avr1 (SIX4) from Fusarium oxysporum f. sp. lycopersici | https://www.rcsb.org/structure/7T6A | RCSB Protein Data Bank, 7T6A |
| Yu DS, Outram MA, Ericsson DJ, Jones DA, Williams SJ | 2023 | Crystal structure of Avr3 (SIX1) from Fusarium oxysporum f. sp. lycopersici | https://www.rcsb.org/structure/7T69 | RCSB Protein Data Bank, 7T69 |
| DS Yu, Ericsson DJ, Williams SJ | 2023 | Crystal structure of SIX6 from Fusarium oxysporum f. sp. lycopersici | https://www.rcsb.org/structure/8EBB | RCSB Protein Data Bank, 8EBB |
| Yu DS, Ericsson DJ, Williams SJ | 2023 | Crystal structure of SIX8 from Fusarium oxysporum f. sp. lycopersici | https://www.rcsb.org/structure/8EB9 | RCSB Protein Data Bank, 8EB9 |

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
