## [Editor Report · eLife assessment]

This study provides **important** new insights into the structural diversity of effectors – proteins secreted by pathogens and symbionts into host cells – from the plant-associated fungus *Fusarium oxysporum* f. sp. *lycopersici*. The study provides a **convincing** approach to elucidate how effectors navigate their host environment by exploiting both computational and experimental approaches to understand how their structure influences binding partners. The work will be of interest to those studying molecular host–microbe interactions and disease protection.

---

## [Referee Report · Reviewer #1 (Public Review)]

Yu et al. investigated Fusarium oxysporum f. sp. lycopersici SIX effectors structure using experimental and computational approaches, and while doing so, the authors identified several SIX effectors as member of the FOLD family, and expanded the known diversity of the SIX effectors. A very interesting and novel finding is the presence of FOLD putative effectors in other Ascomycetes secretome, sharing structural similarities with SIX effectors Avr1, Avr3 and SIX6.

By performing technically sound predictions and experimental confirmation, the authors also confirmed co-operative interactions between Fol effectors, something that was previously known for different pairs of proteins, expanding this observation for new SIX effectors. In addition, the authors contributed to the understanding of the interaction Fol effectors, specifically FOLD and LARS effectors, - I receptors to suppress immunity by structurally similar effectors.

The conclusions of this paper are supported by data and I think it is a pioneer study analyzing the correspondence between AlphaFold predictions and experimentally derived structures, highlighting that models can answer the scientific questions in some cases but could not be enough in others.

---

## [Referee Report · Reviewer #2 (Public Review)]

Yu et al. investigated the structural landscape of 'secreted in xylem' (SIX) effector (virulence and avirulence) proteins from the plant-pathogenic fungus, Fusarium oxysporum f. sp. lycopersici (Fol), with the goal of better understanding effector function and recognition by host (tomato) immune receptors. In recent years, several experimental and computational studies have shown that many effector proteins of plant-associated fungi can be assigned to one of a few major structural families. In the study by Yu et al., X-ray crystallography was used to show that two avirulence effectors of Fol, Avr1 (SIX4) and Avr3 (SIX1), which are recognized by the tomato immune receptors I and I-3, respectively, form part of a new structural family, the Fol dual-domain (FOLD) family, found across three fungal divisions. Using AlphaFold2, an ab initio structural prediction tool, the authors then predicted the structures of all proteins within the Fol SIX effector repertoire (and other effector candidates) and provided evidence that two other effectors, SIX6 and SIX13, also belong to this family.

In addition to identifying members of the FOLD family, structural prediction revealed that proteins of the Fol effector repertoire can largely be classified into a reduced set of structural families. Examples included four members of the ToxA-like family (including Avr2 (SIX3) and SIX8), as well as four members of a new family, Family 4 (including SIX5 and PSL1). Given previous studies had demonstrated that Avr2 (ToxA-like) and SIX5 (Family 4) interact and function together, and that the genes encoding these proteins are divergently transcribed, and because homologues of SIX8 (ToxA-like) and PSL1 (Family 4) from another Fusarium pathogen are functionally dependent on each other and, in the case of Fol, are encoded by genes that are next to each other in the genome, the authors hypothesized that SIX8 and PSL1 may also physically interact. In line with this, co-incubation of the SIX8 and PSL1 proteins, followed by size exclusion chromatography (SEC), gave elution and gel migration profiles consistent with interaction in the form of a heterodimer. AlphaFold2-Multimer modelling then suggested that this interaction was mediated through an intermolecular disulfide bond. Such a prediction was subsequently confirmed through mutational analysis of the relevant cysteine residue in each protein in conjunction with SEC.

Finally, using a variant (homologue) of Avr1 from another Fusarium pathogen, as well as chimeric forms of this protein that integrated regions of Avr1 from Fol, Yu et al. determined through co-expression assays in Nicotiana benthamiana with the I immune receptor, as well as subsequent ion leakage assays, that the C-domain of Avr1 is recognized by the I immune receptor. Furthermore, through these assays, the authors were also able to show that surface-exposed residues in the C-domain enable Avr1 to evade recognition by a variant of the I receptor in Moneymaker tomato that does not provide resistance to Fol.

Overall, the manuscript presents a large body of work that is well supported by the data. A key strength of the manuscript is the validation (benchmarking) of protein structures predicted using AlphaFold2, which is a first for large-scale effector structure prediction papers published to date. Another key strength is the use of large-scale effector structure predictions to make hypotheses about functional relationships or interactions that are then tested (i.e. the SIX8-PSL1 protein interaction and recognition of Avr1 by the I immune receptor). This testing again goes above and beyond the large-scale effector structure prediction papers published to date. Taken together, this showcases how experimental and computational experiments can be effectively combined to provide biologically relevant data for the plant protection and molecular plant-microbe interactions fields.

In terms of weaknesses, the manuscript could have validated the SIX8-PSL1 protein interaction with in planta experiments, such as co-immunoprecipitation assays or co-localization experiments in conjunction with confocal microscopy, to provide support for the interaction in a plant setting. However, given what is already known about the Avr2-SIX5 interaction, these additional experiments are not crucial and could instead form part of a follow-up study.

---

## [Referee Report · Reviewer #3 (Public Review)]

In this work, the authors shed light onto the structures of Fusarium oxysporum f.sp. lycopersici proteins involved in the infection of tomato. They unravelled several new secreted effector protein structures and additionally used computational approaches to structurally classify the remaining effectors known from this pathogen. Through this they uncovered a new and unique structural class of proteins which they found to be present and widely distributed in fungal plant pathogens and plant symbiotic fungi. The authors further predicted structural models for the full SIX effector set revealing that genome-proximal effector pairs share similar structural classes. Building on their Avr1 structure, the authors also define the C-terminal domain and specific amino acid residues that are essential to Avr1 detection by its cognate immune receptor.

A major strength of this work is a portfolio of several (Avr1, Avr3, SIX6, SIX8) new structurally resolved proteins which led to the discovery that several of them fall into the same structural class. These findings are supported by strong results.

The experiments addressing the structure-function relationship of Avr1's avirulence activity are highly relevant to our understanding of disease resistance mechanisms against Fusarium. Additional controls would allow for better support of the conclusions to be drawn. An example is FonSIX4's cell death activity in N.benthamiana leaves and whether FonSIX4 cdll death is indeed dependent on the tomato I receptor. Complementary work in Fusarium mutants lacking Avr1 and expressing chimeric versions would document that the observations from transient expressions in Nicotiana benthamiana are relevant in the biological context of a Fusarium/tomato interaction.

The discovered solvent-exposed residues conditioning Avr1 recognition by the I receptor seem to be positioned in an area of the protein which had previously been highlighted as being highly variable in FOLD proteins of symbiotic fungi but it is not clear from the work whether this is indeed the case or whether Avr1 differs significantly in its structure from FOLD proteins found in other fungi.

It remains to be tested whether the residues conditioning avirulence activity are also crucial for virulence activity in Fusarium.

This work uncovered a new structural class of proteins with critical roles in plant-pathogen interactions. Structure-based predictions and genome-wide comparisons have emerged as a new approach enabling the identification of similar proteins with divergent sequences. The work undertaken by the authors adds to a growing body of work in plant-microbe research, predominantly from pathogenic organisms, and more recently in symbiotic fungi.

---

## [Author Response]

The following is the authors’ response to the current reviews.

We would firstly like to thank all reviewers for their comments and support of this manuscript.

**Reviewer #1 (Recommendations For The Authors):**

No further recommendations.

**Reviewer #2 (Recommendations For The Authors):**

All of my comments have been sufficiently addressed.

**Reviewer #3 (Recommendations For The Authors):**

Thanks for responding to my former recommendations constructively. I believe these points have been fully addressed in this new version.

However, I have not seen any comments on the points I raised in my former public review concerning the I-2 dependence of the FonSIX4 cell death. Do you know whether FonSIX4 would trigger cell death in tissues not expressing any I-2?

We are a little confused concerning this comment. I-2 is a different class of resistance protein (NLR) that recognises Avr2 and this is likely to be intracellular. From the previous public review, we believe reviewer 3 may have been asking us to clarify the dependence of I (MM or M82) on FonSIX4 cell death. We have performed these controls by expressing FonSIX4 and associated FonSIX4/Avr1 chimeras in N. benthamiana (with the PR-1 signal peptide for efficient secretion of effectors) and it does not cause cell death in the absence of the I receptor – see S11F Fig. This was not explicitly conveyed in text so we have included the following in text: “Using the N. benthamiana assay we show FonSIX4 is recognised by I receptors from both cultivars (IM82 and iMoneymaker) and cell death is dependent on the presence of IM82 or iMoneymaker (Fig 5B, S11 Fig).”

I still recommend discussing whether the Avr1 residues crucial for Avr activity are in the same structural regions of the C-terminal domain where previous work has identified residues under diversifying selection in symbiotic fungal FOLD proteins.

The region important for recognition does encompass some residues within the structural region identified to be under diversifying selection in FOLD effectors from Rhizophagus irregularis previously reported (two residues within one beta-strand). However, we also see residues that don’t overlap to this area. We also note that the mycFOLD proteins analysed in symbiotic fungi are heavily skewed towards strong structurally similarity with FolSIX6 (similar cysteine spacing within both N and C-domains and structural orientation of the N and C-domains) rather than Avr1. We are under the impression that Avr1 was not included in the analysis of diversifying selection in symbiotic fungal FOLD proteins, it also is unclear to us if close Avr1 homologues are present. With this in mind, and considering our already lengthy discussion (as previously highlighted during reviewer), we have decided not to include further discussion concerning this point.

The following is the authors’ response to the original reviews.

We would like to thank the editor(s) and reviewers for their work concerning our manuscript. Most of the suggested changes were related to text changes which we have incorporated into the revised version. Please find our response to reviewers below.

**Reviewer #1 (Recommendations For The Authors):**
I only have very minor suggestions for the authors. The first one comes from reading the manuscript and finding it very dense with so many acronyms. This will limit the audience that will read the study and appreciate its impact. This is more noticeable in the Results, with many passages that I would suggest moving to Methodology.

We thank reviewer 1 for their very positive review. We understand that due to the nature of this study, which includes many protein alleles/mutations that were expressed with different boundaries etc., it is difficult to achieve this. Reviewer 2 asked for more details to be provided. We hope we have achieved a nice balance in the revised manuscript.

Something else that would facilitate the reading of the manuscript is the effectors name. The authors use the SIX name or the Avr name for some effectors and it makes it difficult to follow up.

We have tried to make this consistent for Avr1 (SIX4), Avr2 (SIX3) and Avr3 (SIX1). Other SIX effectors are not known Avrs so the SIX names were used.

Reading the manuscript and seeing how in most of the sections the authors used a computational approach followed by an experimental approach, I wonder why Alphafold2-multimer was not used to investigate the interaction between the effector and the receptor?

This is a great suggestion, we have certainly investigated this, however to date there is no experimental evidence to directly support the direct interaction between I and Avr1. Post review, we spent some time trying to capture an interaction using a co-immunoprecipitation approach however to date we have not been able to obtain robust data that support this. We are currently looking to study this utilising protein biophysics/biochemistry but this work will take some time.

**Reviewer #2 (Recommendations For The Authors):**

We thank reviewer 2 for the very thorough editing and recommendations. We have incorporated all minor text edits below into the manuscript.

Line 43: perhaps "Effector recognition" instead of "Effector detection", to be consistent with line 51?Line 60: Change to "leads".Line 79: Italicise Avr2.Line 94: Add the acronym ETI in parentheses after "effector-triggered immunity".Line 106: "(Leptosphaeria Avirulence-Supressing)" should be "(Leptosphaeria Avirulence and Supressing)".Line 112: Change "defined" to "define".Line 119: Spell out the species name on first use.Line 205: Glomeromycota is a division rather than a genus. Consistent with Fig 2, it alsodoes not need to italicized.Line 207: Change "basidiomycete" to "Division Basidiomycota", consistent with Fig 2.Line 214: Change "alignment of Avr1, Avr3, SIX6 and SIX13" to "alignment of the mature Avr1, Avr3, SIX6 and SIX13 sequences".Line 324: Change "solved structures" to "solved protein structures".Line 335: Spell out acronyms like "MS" on first use in figure legends. Also dpi in other figure legends.Line 341: replace "effector-triggered immunity (ETI)" with "(ETI)" - see comment on Line 94.Line 370: Change "domains" to "domain".Line 374: In the title, change "C-terminus" to C-domain", consistent with the rest of the figure legend.Line 404: Change "(basidiomycetes and ascomycetes)" to "(Basidiomycota and Ascomycota fungi)", consistent with Fig 2C.Line 416: Change "in" to "by".Line 427: un-italicize the parentheses.Line 519: First mention of NLR. Spell out the acronym on first use in main text.S5 and S11 figure titles should be bolded.Line 852: Replace "@" with "at".S4 Table: Gene names should be italicised.S5 Table: Needs to be indicated that the primer sequences are in the 5´-3´ orientation.With regards to the Agrobacterium tumefaciens-mediated transient expression assays involving co-expression of the Avr1 effector and I immune receptor, the authors need to make clear how many biological replicates were performed as this information is only provided for the ion leakage assay.

We have added these data to the figure legend

Line 57: For me, the text "Fol secretes a limited number of structurally related effectors" reads as Fol secretes structurally related effectors, but very few of them are structurally related. Perhaps it would be better to say that the effector repertoire of Fol is made up of proteins that adopt a limited number of structural folds, or that the effector repertoire can be classified into a reduced set of structural families?

This edit has been incorporated.

Lines 66-67: Subtle re-wording required for "The best-characterized pathosystem is F. oxysporum f. sp. lycopersici (Fol)", as a pathosystem is made up of a pathogen and its host. Perhaps "The best-characterized pathosystem involves F. oxysporum f. sp. lycopersici (Fol) and tomato".

Sentence has been reworded.

Line 113 and throughout: Stick with one of "resistance protein", "receptor", "immune receptor" and "immunity receptor" throughout the manuscript.

We have decided to use both receptor and immunity receptor as not all receptors investigated in the manuscript provide immunity.

Lines 149-150: The title does not fully represent what is shown in the figure. The text "that is unique among fungal effectors" can be deleted as there is nothing in Fig 1 that shows that the fold is unique to fungal effectors.

Figure title has been changed.

Line 173: The RMSD of Avr3 is stated as being 3.7 Å, but in S3 Fig it is stated as being 3.6 Å.

This was a mistake in the main text and has been corrected.

Lines 202-204: This sentence needs to be reworded, as the way that it is written implies that the Diversispora and Rhizophagus genera are in the Ascomycota division. Also, "Ascomycetes" should be changed to "Ascomycota fungi", consistent with Fig 2.

Sentence has been reworded.

Line 233: "Scores above 8". What type of scores? Z-scores?

These are Z-scores. This has been added in text.

Lines 242-246: It is stated that SIX9 and SIX11 share structural similarity to various RNA-binding proteins, but no scores used to make these assessments is given. The scores should be provided in the text.

Z-scores have been added.

Fig 4A: SIX3 should be Avr2, consistent with line 292. The gene names should be italicised in Fig 4A.

SIX3 was changed to Avr2. Gene names have been italicised.

Line 356: Subtle rewording required, as "co-infiltrated with both IM82 and iMoneymaker" implies that you infiltrated with protein rather than Agrobacterium strains.

Sentence has been reworded.

Fig 5A, Fig 5C and Line 380: Light blue is used, but this looks grey. Perhaps change colour, as grey is already used to show the pro-domain in Fig 5A (or simply change the colour used to highlight the pro-domain)?

Colour depicting the C-domain was changed.

Lines 530-531: This text is no longer correct. Rlm4 and Rlm3 are now known to be alleles of Rlm9. See: Haddadi, P., Larkan, N. J., Van deWouw, A., Zhang, Y., Neik, T. X., Beynon, E., ... & Borhan, M. H. (2022). Brassica napus genes Rlm4 and Rlm7, conferring resistance to Leptosphaeria maculans, are alleles of the Rlm9 wall‐associated kinase‐like resistance locus. Plant Biotechnology Journal, 20(7), 1229.

We thank the reviewer for picking this up. This text has been updated.

Line 553: Provide more information on what the PR1 signal peptide is.

More information about the PR1 signal peptide has been added.

Lines 767-781: Descriptions and naming conventions of proteins throughout the figure legend need to be consistent and better reflect their makeup. For example, I think it would be best to put the sequence range after each protein mentioned - e.g. Avr118-242 or Avr159-242 instead of Avr1, PSL1_C37S18-111 instead of PSL1_C37S, etc. Furthermore, it is often stated that a protein is full-length when it lacks a signal peptide - my thought is that if a proteins lack its signal peptide, it is not full-length. The acronym "PD" also needs to be spelled out as "pro-domain (PD)" in the figure legend.

We have incorporated sequence range for proteins that were produced upon first use. Sequence ranges that were modelled in AlphaFold2 were not added in text because they can be found in Supplementary Table 3.

Lines 853-845: It is stated the sizes of proteins are indicated above the chromatogram in S10 Fig, but this is not the case. It is also not clear from S10B Fig that the faint peaks correspond to the peaks in the Fig 4B chromatogram. In S10D Fig, the stick of C58S is difficult to see. Perhaps change the colour or use an arrow/asterisk?

Protein size estimates have been added above the chromatogram. Added text to indicate that the faint peaks correspond to peaks in Fig 4B. Added an asterisk in S10D Fig to identify the location of C58.

S14 Fig is not mentioned/referenced in the main text of the manuscript.

This was a mistake and has been added.

The reference list needs to be updated to accommodate those referenced bioRxiv preprints that have now been published in peer-reviewed journals.

The reference list has been updated.

**Reviewer #3 (Recommendations For The Authors):**
It would be good to discuss whether the pro-domains affecting virulence or avirulence activity.

Kex2, the protease that cleaves the pro-domain functions in the golgi. We therefore suspect that the pro-domain is removed prior to secretion. For recombinant protein production in *E. coli* we find that these pro-domains are necessary to obtain soluble protein (doi: 10.1111/nph.17516). As we require the pro-domain for protein production and can not completely removing them from our preps, we cannot perform experiments to test this and subsequently comment further. In a paper that identified SIX effectors in tomato utilising proteomics approach (available here), it appears that the pro-domains were not captured in this analysis. This supports the conclusion that they are not associated with the mature/secreted protein.

The authors stated that the C-terminal domain of SIX6 has a single disulfide bond unique to SIX6. Please clarify in which context is it unique: in Fusarium or across all FOLD proteins?

This is in direct comparison to Avr1 and Avr3. The disulfide in the C-domain of SIX6 is unique compared to Avr1 and Avr3. This has been made clear in text.

The structural similarity of FOLD proteins to other known structures have been discussed (lines 460ff), but it is not clear whether all structures and models identified in this work would yield cysteine inhibitor and tumor necrosis factors as best structural matches in the database or whether this is specific to a single FOLD protein. Please consider discussing recently published findings by others (Teulet et al. 2023, New Phytologist) on this aspect.

This analysis was performed for Avr1, we obtained relatively low similarity hits for Avr3/Six6. We have updated this text accordingly… “Unfortunately, the FOLD effectors share little overall structural similarity with known structures in the PDB outside of the similarity with each other. At a domain level, the N-domain of the FOLD effector Avr1 has some structural similarities with cystatin cysteine protease inhibitors (PDB code: 4N6V, PDB code: 5ZC1) [60, 61], and the C-domain with tumour necrosis factors (PDB code: 6X83) [62] and carbohydrate-binding lectins (PDB code: 2WQ4) [63]. Relatively weak hits were observed for Avr3/Six6.”

It might be useful to clearly point out that the ToxA fold and the C-terminus of the FOLD fold are different.

We have secondary structural topology maps of the FOLD and ToxA-like families in S8 Fig which highlight the differences in topology between these two families.

Please add information to Fig.S8 listing the approach to generate the secondary structure topology maps.

We have added this information in the figure caption.